# A new snow depth data set over northern China derived using GNSS interferometric reflectometry from a continuously operating network (GSnow-CHINA v1.0, 2013-2022)

**Wei Wan[1], Jie Zhang[2], Liyun Dai[3], Hong Liang[4], Ting Yang[5], Baojian Liu[1], Zhizhou Guo[1], Heng Hu[4], Limin Zhao[6]**

[1] Institute of Remote Sensing and GIS, School of Earth and Space Sciences, Peking University, Beijing 100871,

China

[2] College of Oceanography and Space Informatics, China University of Petroleum (East China), Qingdao 266580,
China

[3] Key Laboratory of Remote Sensing of Gansu Province, Northwest Institute of Eco-Environment and Resources,
Chinese Academy of Sciences, Lanzhou 730000, China

[4] Meteorological Observation Center, China Meteorological Administration, Beijing 100081, China

[5] Institute of Geographic Sciences and Natural Resources Research, Chinese Academy of Sciences, Beijing, China

[6] Aerospace Information Research Institute, Chinese Academy of Sciences, Beijing 100101, China

**Correspondences to**: Wei Wan (w.wan@pku.edu.cn) and Hong Liang (liangh@cma.gov.cn)

**Abstract:** The currently available long-term snow depth data sets are either from point-scale ground measurements or from gridded satellite/modeled/reanalysis data with coarse spatial resolution, which limits the applications in climate model, hydrological model, and regional snow disaster monitoring. Benefit from its unique advantages of cost-effective and high spatial-temporal resolution ($\sim 1000$ m$^2$, hourly in theory), snow depth retrieval using the Global Navigation Satellite System Interferometric Reflectometry (GNSS-IR) technique has become a popular topic in recent years. However, due to complex environmental and observation conditions, developing robust and operational technology to produce long-term snow depth data sets using observations from various GNSS station networks is still challenging. The two objectives of this study are 1) to propose a comprehensive framework using raw data of the complex GNSS station networks to retrieve snow depth and to control its quality automatically; and 2) to produce a long-term snow depth data set over northern China (i.e., GSnow-CHINA v1.0, 12h/24h, 2013-2022) using the proposed framework and historical data from 80 stations. The data set has high internal consistency with regards to different GNSS constellations (mean r = 0.98, RMSD = 0.99 cm, and nRMSD (snow depth > 5 cm) = 0.11), different frequency bands (mean r = 0.97, RMSD = 1.46 cm, and nRMSD (snow depth > 5 cm) =0.16), and different GNSS receivers (mean r = 0.62). The data set also has high external consistency with the in-situ measurements and the passive microwave (PMW) product, with a consistent illustration of the interannual snow depth variability. The results also show the potential of GNSS to derive hourly snow depth observations for better monitoring snow disasters. The proposed framework to develop the data set provides comprehensive and supportive information for users to process raw data of ground GNSS stations with complex environmental conditions and various observation conditions. The resulting GSnow-CHINA v1.0 data set is distinguished from the current point-scale in-situ data or coarse-gridded data, which can be used as an independent data source for validation purposes. The data set is also useful for regional climate research and other meteorological and hydrological applications. The algorithm and the data files will be maintained and updated as more data become available in the future. The GSnow-CHINA v1.0 data set is available at National Tibetan Plateau/Third Pole Environment Data Center via https://doi.org/10.11888/Cryos.tpdc.271839 (Wan et al. 2021).

**Keywords**: Snow depth, Global Navigation Satellite System interferometric reflectometry (GNSS-IR), GNSS station networks, northern China

50

## 1 Introduction

Snow cover is one of the most active elements in the cryosphere, and the maximum snow area during winter nearly occupies 50% of the total land surface area of the Northern Hemisphere (Frei and Robinson, 1999; Armstrong and Brodzik, 2001; Robinson et al., 1993). The snow change plays a significant role in the hydrological, ecological, and climatic systems (Henderson et al., 2018). Therefore, accurately estimating snow cover and snow depth and their variations is essential for studies on climate and hydrology.

Currently, snow cover products derived from optical remote sensing data present high accuracy (Hao et al., 2021), but snow depth products show significant uncertainties. Snow depth can be measured at point-scale using ground-based ultrasonic snow depth sensors or laser snow depth sensors, and mainly include observations from meteorological stations, snow surveys, and hydrological stations (Kinar and Pomeroy, 2015). Large-scale snow depth can be retrieved from optical, passive microwave, and active remote sensing observations (Shi and Dozier, 2000; Guerreiro et al., 2016; Leinss et al., 2014; Che et al., 2016), yet currently operational observations have shortcomings. Optical remote sensing is affected by solar radiation and cloud (Dai et al., 2017). Passive microwave remote sensing is with coarse spatial footprints (> 25 km), and the observations saturate in deep snow (> 0.8 m) (Lievens et al., 2019). Active microwave remote sensing has a long revisiting period (> 20 days) and high cost (Lievens et al., 2019).

The available global/hemispheric/regional snow depth data sets are mainly derived from ground observations, microwave remote sensing, model simulations, and reanalysis (Xiao et al., 2020). Representative snow depth data sets include: 1) In-situ measurements from ground networks such as SCAN and SNOTEL in the United States (point-scale, hourly/daily/7-day/monthly; http://www.wcc.nrcs.usda.gov), 2) Data sets derived from satellite passive microwave brightness temperatures, e.g., the Advanced Microwave Scanning Radiometer for the Earth Observing System (AMSR-E) and its follow-on the Advanced Microwave Scanning Radiometer-2 (AMSR2) (25 km, daily, global/regional, 2002-, https://nsidc.org/), and the Global Snow Monitoring for Climate Research (GlobSnow) data set produced from the data assimilation of microwave radiometer data and meteorological station data (25 km, daily, hemispheric, 1979-, https://www.globsnow.info/), 3) Snow depth data set simulated using model such as snow modules in the Global Land Data Assimilation System (GLDAS-2.0, 1948, 0.25° x 0.67°, 3-hourly and monthly; https://ldas.gsfc.nasa.gov/gldas), and 4) Reanalysis snow depth data

sets from the ERA-Interim (1979-, 0.75°, 6-hourly/daily/monthly; http://www.ecmwf.int/) and the Modern-Era Retrospective Analysis for Research and Applications as well as their series data sets ((MERRA/MERRA-2/MERRA-Land, 1979, 0.5° x 0.67°; https://gmao.gsfc.nasa.gov/reanalysis/).

The aforementioned long-term snow depth data sets are either point-scale or gridded data with coarse spatial resolution. Previous studies also demonstrated that current snow depth data sets and snow water equivalent data sets show significant inconsistencies and uncertainties, which limit their applications in climate change projections and hydrological processes simulations (Xiao et al., 2020; Zhang et al., 2021; Shao et al., 2022). Due to the complex spatial-temporal variability and the limitations of the current observation approaches, it is still challenging to derive snow depth data set of long-term with high spatial-temporal resolution. In particular, it lacks detailed observations of snow depth on a regional scale, which limits the applications in climate models, hydrological models, and snow disaster monitoring.

Estimating snow depth using the Global Navigation Satellite System Interferometric Reflectometry (GNSS-IR) technique has become a popular topic in recent years, ever since the principle was proposed by (Larson et al., 2009). Snow depth is determined by calculating the relative change of the effective multipath reflector height (i.e., the snow surface) to the snow-free surface. This technique is cost-effective because it does not require an additional transmitter, and instead, it continuously receives L-band microwave signals transmitted by the GNSS satellites. The temporal resolution for snow sensing is expected to be hourly, along with the increasing number of GNSS satellites in orbit (Tabibi et al., 2017a). For typical GNSS-IR sites the spatial footprint is ~1000 $m^2$, which is a scale between point-scale and satellite-scale (i.e., from tens of meters to tens of kilometers) (Larson and Nievinski, 2013). Therefore, GNSS-IR could provide new snow depth data sets which could be supplementary to the current in-situ and satellite data sets. However, developing robust and operational technology to produce long-term snow depth data sets using data from various GNSS station networks is still challenging due to complex environmental and observation conditions.

This study, taking advantage of 80 sites from a continuously operating GNSS network over northern China, develops a comprehensive framework to process raw data from various stations and subsequently develops a new GNSS-IR snow depth data set (GSnow-CHINA v1.0, 12h/24h, 2013-2022). Northern China has a wide-distributed snow cover from October to April next year. China's annual mean snow extent is greater than

9,000,000 km$^2$, with a stable snow-covered area of ~ 4,200,000 km$^2$. This region is the main snow-covered area in China, which also plays a vital role in the climate research of the Northern Hemisphere and the cryosphere. The unique characteristics of the GSnow-CHINA v1.0 and the framework to develop it are as follows:

(1) GSnow-CHINA v1.0 is a snow depth data set developed using GNSS data source, independent from the current satellite, modeled, reanalysis, and in-situ data sets. The spatial resolution of this data set is between the

in-situ point-scale and the coarse-gridded data, which makes it a new data set suitable for validation purposes.

(2) GSnow-CHINA v1.0 is a long-term snow depth data set over China with high temporal and spatial resolution, which provides a new data source for regional and global climate research. The data set is also helpful for monitoring local snow disasters and water resource management.

(3) The proposed framework to develop the data set provides comprehensive and supportive information for

users to process raw data of ground GNSS stations with complex environmental conditions and various observation conditions. The technique has the potential to provide finer resolution snow depth product (e.g., one to two hours) with adequate observations from multiple GNSS systems.

## 2 Study area and data

### 2.1 Study area

Northern China lies between latitudes 25°N and 55°N and longitudes 70°E and 140°E and includes humid, semi-humid, semi-arid, and arid zones. Snow is the primary freshwater resource in this area. Sudden snowstorms or long-lasting deep snow is one of the major natural disasters for pastoral areas because it affects livestock grazing. The study area includes the three main stable snow accumulation areas over China, i.e., Northeast China and Inner Mongolia (NCM), North Xinjiang and Tianshan mountain (NXT), and Qinghai-Tibet Plateau (QTP)

(Figure 1).

The NCM region has various geomorphic types. Mountains and hills surround the east, west, and north of this region, and the middle of this region is plain. The mean minimum air temperature in January is below -30 °C. The annual mean snow depth is greater than 5 cm with a maximum value of greater than 30 cm. The mean snow density of this area is ~0.15 g. cm$^{-3}$. The NXT region has abundant seasonal snow water resources, vital to local

irrigation and animal husbandry. The mean air temperature is -4 ~ 9 °C with a long winter period. The QTP region is the core region of "The Third Pole" with a mean altitude of ~ 4378 m. Rainfall of the QTP is

concentrated chiefly from May to September, while snowfall usually starts from September to April of the following year.

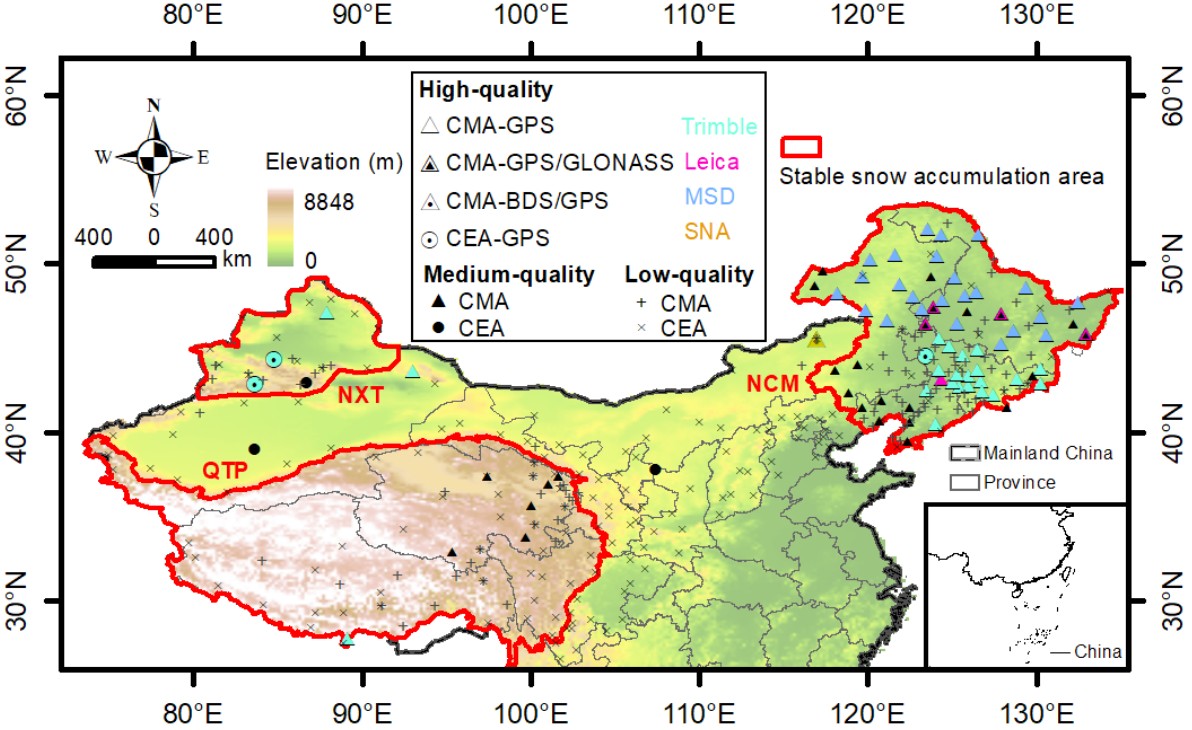

**Figure 1.** Distributions of the GNSS sites over northern China. The symbols are colored by the GNSS receiver type, such as Trimble, Leica, MinShiDa (MSD), and SiNan (SNA).

**2.2 Data**

Observations from the GNSS station networks over northern China are the primary data source to produce the snow depth data set. The networks include two separate categories constructed by two organizations, i.e., the network constructed by the China Meteorological Administration (CMA) and the Crustal Movement Observation Network of China constructed by the China Earthquake Administration (CEA). China started to construct ground GNSS stations in 2009, and the station build phase was initially completed in 2012 with some regions later in 2015. The CMA stations were built to observe precipitable water vapor, while the CEA stations were built to monitor crustal deformation.

As shown in Figure 1, raw data from all 174 CMA sites and 171 CEA sites are acquired from the Center of Meteorological Observation, CMA, to initially evaluate the capability to retrieve snow depth site by site. The

sites are divided into three categories, i.e., high quality, medium quality, and low quality, following the recognition rule used for site quality determination. The rule will be introduced in the following "Methods"

Section. Overall, there are 55 high-quality sites (52 for CMA and 3 for CEA) and 25 medium-quality sites (22 for CMA and 3 for CEA). The high-quality CMA sites are composed of various types regarding the received data of different GNSS systems, i.e., 47 GPS-only, 4 GPS/GLONASS compatible, and 1 GPS/BDS compatible. The CEA sites are GPS-only sites. Most of the high-quality sites are located in the NCM region, while a few are located in the NXT and QTP regions.

155        Figure 2 shows the periods of the high-quality and medium-quality GNSS sites used for snow depth retrieval. For CMA, despite the possible raw data missing for some sites, the majority time spans for the high-quality sites are 2013-2022, 2015-2022, and 2016-2022, and that for the medium-quality sites are 2015-2022. For CEA, the three high-quality sites are from 2016/2018/2019-2022, with one medium-quality site having the earliest record from the year 2010. Each GNSS site has irreplaceable value because of its unique natural environment and

characteristic of snow. Therefore, regardless of the raw data incompleteness in some periods for some sites, we preserve the high-quality and medium-quality sites as much as possible during the production of the data set.

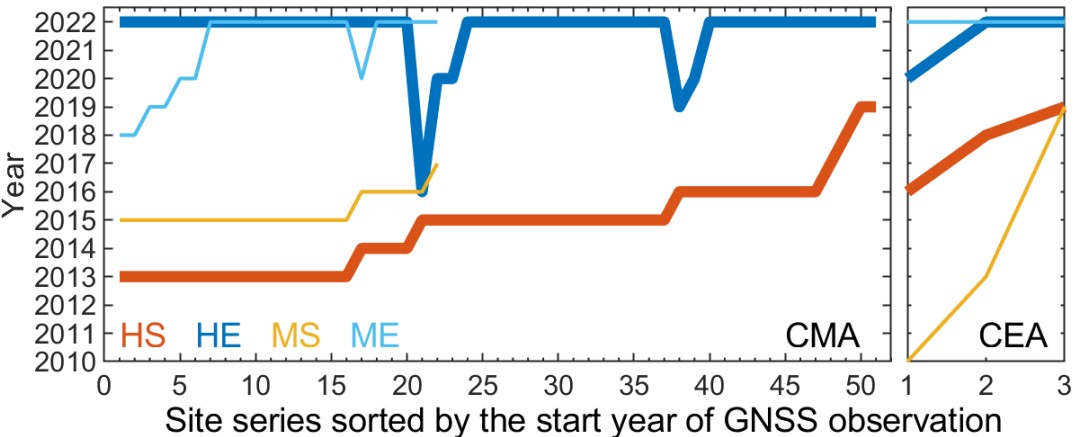

**Figure 2.** Periods of the GNSS sites used for snow depth retrieval. HS: start year of the high-quality site; HE: end year of the high-quality site; MS: start year of the medium-quality site; ME: end year of the medium-quality site.

CMA: China Meteorological Administration; CEA: China Earthquake administration.

The broadcast ephemeris was used to calculate each GNSS satellite's position. For CMA and CEA sites, the minimum elevation angle of the GNSS satellite was set to be 10° when the sites were built.

The Soil Moisture Active and Passive (SMAP) L3 36 km soil moisture data are used to estimate the penetration depth of GNSS signals to the soil layer (O'neill et al., 2019). It is a quality control step to derive a more accurate reflector height of the snow-free surface. The Moderate Resolution Imaging Spectroradiometer (MODIS) 1 km Normalized Difference Vegetation Index (NDVI) data are used to identify the vegetation effects on snow depth retrieval (Didan, 2021). Two independent snow depth data products are used to analyze the quality of the data set produced in this study. One is the 1979-2020 snow depth product using passive microwave remote sensing produced by (Che and Dai, 2015; Che et al., 2008; Dai et al., 2015) (daily, 25 km), named PMW hereafter for short. The snow depth of this product is derived using the SMMR and SSMI/S microwave brightness temperature processed by the National Snow and Ice Data Center (NSIDC). The other is the daily in-situ snow depth measurements using laser snow depth sensors provided by the Meteorological Observation Center, CMA.

**3    Methods**

The flowchart to produce and validate the GSnow-CHINA data set is shown in Figure 3. The raw GNSS data used for snow depth retrieval is the daily Receiver Independent Exchange Format (RINEX) data derived directly from individual CMA/CEA GNSS sites. Significant steps to produce the data set are described as follows:

1) The observables for snow depth retrieval, i.e., satellite Pseudorandom Noise (PRN) numbers, observation time, satellite elevation angle, satellite azimuth angle, pseudorange, carrier phase (CP), and Signal-to-Noise Ratio (SNR), are extracted or calculated from the raw data.

2) The Lomb-Scargle Periodogram (LSP) analysis (Lomb, 1976) is executed on several snow-free days to determine the mean reflector heights for each GNSS satellite, each quadrant, and each GNSS frequency. For those high- and medium-quality sites which will be distinguished in the following Step 3), the mean reflector heights are used as reference heights when calculating snow depth. Here, the reflector height means the vertical distance between the antenna phase center and the surface.

3) A comprehensive evaluation of the quality of all the GNSS sites is done based on the data quality of the snow-free surface reflector heights in Step 2), and the sites are divided into high-, medium-, and low-quality accordingly.

4) For high- and medium-quality sites, the model for deriving daily reflector height is established, and the raw snow depth for each GNSS satellite, each quadrant, and each GNSS frequency is subsequently calculated as the difference value of the referenced height in Step 2) and the height of this step.

5) Several quality-control strategies are used to further improve the quality of the raw snow depth during the previous step, such as considering the penetration depth of soil, considering the vegetation effects, filtering of outliers, adding valid flags such as the standard error (STE) of snow depth and the number of PRNs used to calculate a specific snow depth value.

6) Daily 24-hour and sub-daily 12-hour snow depth are derived for general high- and medium- GNSS sites, and snow depths of finer resolution are additionally derived for several GPS/GLONASS compatible sites.

7) The GSnow-CHINA data set is evaluated using the PMW product and the in-situ measurements. The advantages and limitations of the produced data set are further analyzed to provide supportive information for future method improvement or data set extension.

The following sections introduce detailed descriptions of the solutions of several key steps in the processing framework.

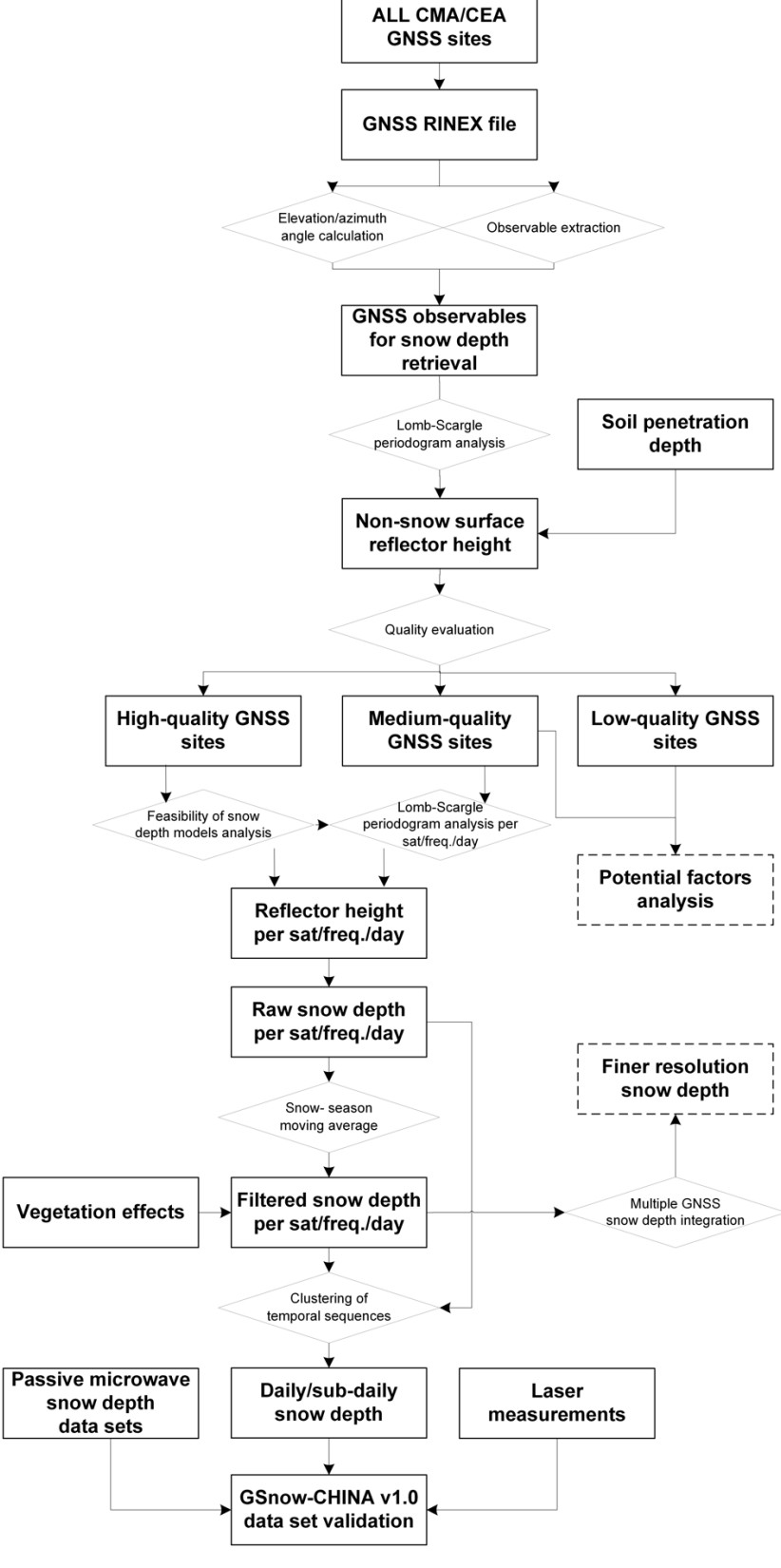

210    **Figure 3.** Flowchart showing the production and validation of the GSnow-CHINA v1.0 data set.

### 3.1 Snow depth retrieval model

The state-of-the-art GNSS-IR snow depth retrieving models can be divided into two categories according to the two types of observables (i.e., the SNR and the carrier phase (CP)). The principle of the SNR model is to establish a linear relationship between the oscillation frequency of the SNR observation sequence of the reflected signal and the height of the reflection surface (Larson et al., 2009). This model was later derived into several variants: e.g., the triple-frequency SNR combination model (SNR_COM) (Zhou et al., 2019), the SNR model based on raw SNR sequences (Peng et al., 2016), the SNR model based on horizontal polarization antenna (Chen et al., 2014), the SNR model considering the influence of construction facilities (Vey et al., 2016), and the SNR model considering the influence of terrain (Zhang et al., 2017). The carrier phase combination model was initially proposed to estimate snow depth when there were no SNR data in the raw GNSS observation file (Ozeki and Heki, 2012). The initial form of this model was using the geometry-free linear combinations of the phase measurements (L4), and (Yu et al., 2015; Yu et al., 2018) extended the model to use triple-frequency carrier phase observations (F3) as well as the combination of pseudorange and carrier phase of dual-frequency signals (F2C).

The main formulas and applicability of the five models mentioned above to the data of GNSS sites in this study are listed in Table 1, and Table 2 further shows the meanings of variables for the models in Table 1. The SNR, L4, and F2C models are suitable for all sites because the observables used as inputs for these models are available in the GNSS raw data. The SNR model has been verified to have higher accuracy than the L4 and F2C models (Liu et al., 2022). The applicability of the SNR_COM and F3 models is limited because most of the GNSS sites do not contain three SNR or CP observables in a single raw data file. Considering both the applicability and the accuracy, the SNR model is determined as the primary model used to produce the snow depth data set.

**Table 1** Snow depth models and their corresponding formulas

| Model | Main formulas | Applicability |
|---|---|---|
| **SNR (Larson et al., 2009)** | $$\mathrm{SNR}^2 = A_c^2 = A_d^2 + A_m^2 + 2A_dA_m \cos Q$$ $$A_m = A\cos\left(\frac{4\pi h}{\lambda}\sin E + \varphi\right)$$ $$f = \frac{2h}{\lambda}$$ | **Suitable for all sites** |
| SNR_COM (Zhou et al., 2019) | $$SNR_{com,i} = [SNR_{1,i}\,SNR_{2,i}\,SNR_{3,i}]$$ | Only suitable for several BDS sites (no triple SNR observations) |
| L4 (Ozeki and Heki, 2012) | $$L_1 = \rho + I(f_1) + T + M_{L1} + noise_1$$ $$L_2 = \rho + I(f_2) + T + M_{L2} + noise_2$$ $$L_4 = L_1 - L_2 = I(f_1) - I(f_2) + M_{L1} - M_{L2} + noise_1 - noise_2$$ | Suitable for all sites but with relatively lower accuracy |
| F3 (Yu et al., 2015) | $$L_3 = \rho + I(f_3) + T + M_{L3} + noise_3$$ $$f_3 = \lambda_3^2(L_1 - L_2) - \lambda_2^2(L_1 - L_3) + \lambda_1^2(L_2 - L_3)$$ | Suitable for one GPS/GLONASS site |
| F2C (Yu et al., 2018) | $$c_1 = \rho + I(f_1) + T + M_{c1}$$ $$f_{2c} = \frac{\lambda_1^2 + \lambda_2^2}{\lambda_1^2 - \lambda_2^2}(c_1 - L_1) - \frac{2\lambda_1^2}{\lambda_1^2 - \lambda_2^2}(c_1 - L_2)$$ | Suitable for all sites but with relatively lower accuracy |

235 **Table 2** Meanings of variables for the models in Table 1

| Variables | Meanings |
|---|---|
| $A_d$ | Amplitudes of the direct signal |
| $A_m$ | Amplitudes of the reflected signal |
| $A_c$ | Amplitudes of the synthetic signal |
| $\cos Q$ | Cosine value of the angle between the direct signal and the reflected signal |
| $\lambda$ | Carrier wavelength |
| $E$ | Satellite elevation angle |
| $h$ | Vertical reflection distance |
| $f$ | Frequency of GNSS multipath reflection signal |
| $\varphi$ | Phase values less than an entire period |
| $SNR_{com,i}$ | SNR observation values of triple-frequency |
| $\lambda_i$ | Wavelength |
| $\rho$ | The true geometric range between the satellite and receiver |
| $T$ | Tropospheric delay |
| $I(f_i)$ | Ionospheric delay for the signal |
| $M_{Li}$ | Multipath error for the signal |
| $noise_i$ | Integer ambiguities for the signal |
| $L_4$ | Multipath error sequence of L4 |
| $f_3$ | Multipath error sequence of F3 |
| $f_{2c}$ | Multipath error sequence of F2C |

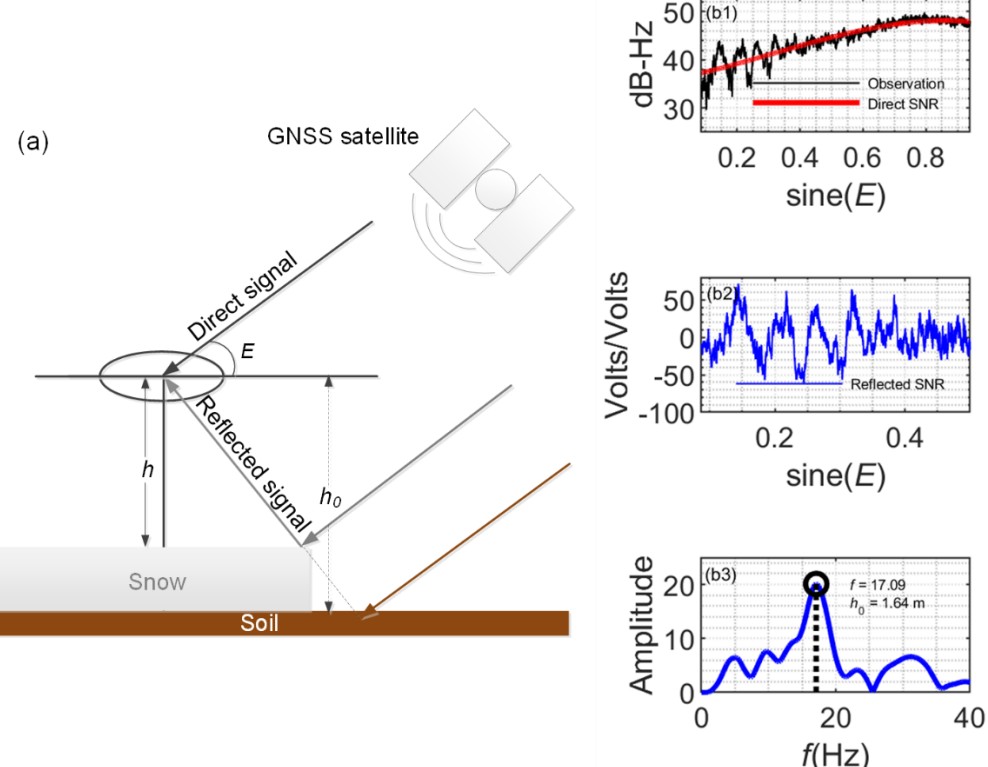

**Figure 4.** Geometry and principle of the SNR model. (a) The geometry of the direct and reflected signal over the snow surface; (b1) Example of the recorded GNSS SNR data and the removal of the direct signal with a second-order polynomial; (b2) Residual of (b1) below elevation angle ($E$) of 30°, converted from dB to linear units (for simplicity, Volts); (b3) Lomb-Scargle analysis of (b2) to find out the dominant frequency of the transformation and the resulting reflector height.

The geometry and principle of the SNR model are shown in Figure 4. As shown in Figure 4a, the snow depth ($h_{snow}$) is calculated using a simple equation:

$$h_{snow} = h_0 - h \tag{1}$$

Where $h_0$ is the reflector height of the snow-free surface, and $h$ is the reflector height of the snow-covered surface. The approaches to derive $h_0$ and $h$ are similar, with Figure 4 b1~b3 showing the general technical process. First, the time series of GNSS SNR observation are shown as a function of sine (elevation angle), and the direct signal is removed using the polynomial fitting method. The remaining is treated to be the contribution of the reflected signal from the land surface. Second, the reflected signal is converted from dB-Hz

to Volts/Volts. Third, the LSP analysis is executed to the reflected signal curve to find out the dominant frequency of the transformation. In this study, the Peak-to-Noise Ratio (PNR) of the LSP is set to be greater than 5 to filter out the quality‑controlled satellite tracks. The $h_0$ or $h$ can be calculated by (Larson et al., 2009):

$$h = \lambda f / 2 \tag{2}$$

Where $\lambda$ is the wavelength of the GNSS signal and $f$ is the dominant frequency.

**3.2 Determination of the snow-free surface reflector height**

For each site, ~ ten days of data with no snow on the ground are used to calculate the raw snow-free surface reflector height ($h_0$). According to the data availability, days of the year (DOYs) 110~119 or DOYs 274~283 are generally selected since these days have no snow according to historical in-situ data. Specifically, for GLONASS, to deal with the non-repeating tracks, one-month snow-free data (DOY 105~135) are used to calculate the raw $h_0$. The reflector height for each GNSS satellite, quadrant, and GNSS frequency band is calculated using the Lomb-Scargle spectrum, and it is just the initial height being used for the quality evaluation of the GNSS sites. Due to the complex natural environment for various sites, it is not clear whether one site is suitable for snow depth retrieval. The following section will define a rigorous rule to evaluate the quality of all the GNSS sites. For those high- and medium-quality sites determined in the following section which are suitable for snow depth retrieval, the finalized snow-free surface reflector height will be determined as the mean value of heights of the ten days.

It is worth mentioning that GPS ground tracks have sidereal repeatability and reappear at the same azimuth every day. In contrast, GLONASS satellite and BDS MEO satellite have non-repeating ground tracks. GLONASS orbits repeat every eight sidereal days, with the ground track shifted by 45° in longitude per day (Tabibi et al., 2017b). BDS MEO satellites repeat approximately every seven sidereal days (Ye et al., 2015). In this study, there are only 4 GLONASS sites (i.e., bfqe, bttl, hltl, and hlhl) and 1 BDS site (e.g., qxdw). The strategy for processing GLONASS data is slightly different from that of GPS, i.e., the snow-free surface reflector heights are given in twelve azimuths separated by 30° for all available GLONASS satellite tracks and frequency bands. While for BDS satellite, due to the relatively low number of available satellites, the reflector height is given by quadrant only without distinguishing tracks and frequency bands to preserve as many observations as possible. Previous research developed a multistep clustering algorithm to handle the non-repeating ground tracks of GLONASS

(Tabibi et al., 2017a). We are also developing a new algorithm in an upcoming study considering terrain effects,

which will be particularly effective for non-repeating tracks.

### 3.3 Quality evaluation of the GNSS sites

The CMA and CEA sites are built under various natural and manual environmental conditions. Figure 5 shows several photos of typical CMA/CEA sites. The CMA sites are mainly built on the ground with antenna height ranging from 1.5 m to 5 m. Some sites are located in relatively flat and open land, while others are in yards

and are surrounded by buildings or other artificial objects. The majority of the CEA antennas are set upon a rooftop (e.g. Site "qhdl" in Figure 5), with the GNSS receivers being put in the accompanying housing. It explains why most of the CEA sites are not suitable for snow depth retrieval.

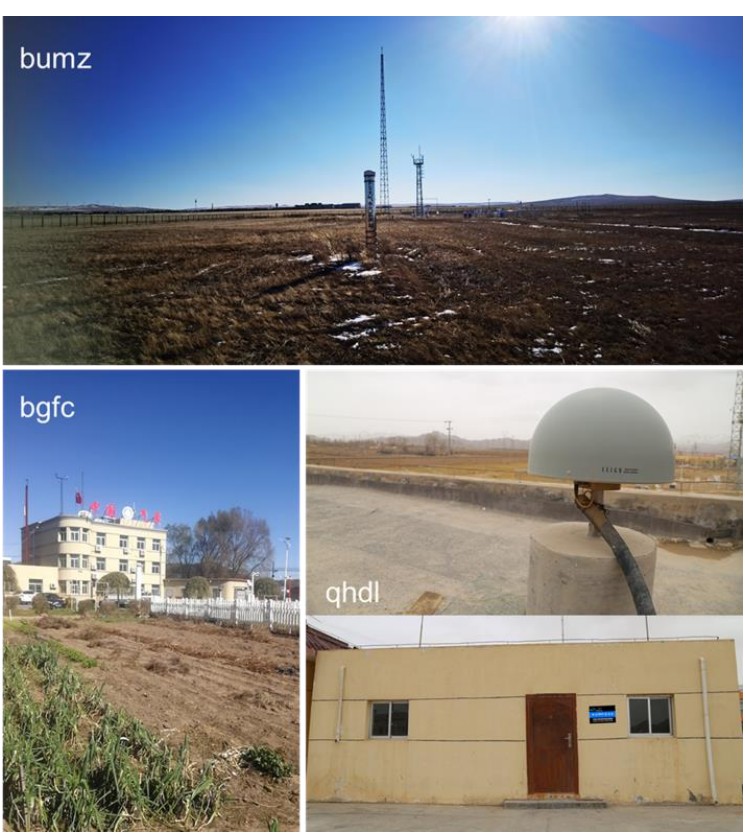

**Figure 5.** Photos of typical GNSS sites. "bumz" and "bgfc" are two high-quality CMA sites, and "qhdl" is a low-

quality CEA site that is not suitable for snow depth retrieval.

A rigorous rule is defined to evaluate the quality of all the GNSS sites. For each site, the 10-day reflector heights of snow-free surface (i.e., $h_0$) are calculated, sorted, and colored by azimuths to make a "$h_0$ plot".

Examples of the "$h_0$ plot" are shown at the bottom of each subfigure in Figure 6. The "$h_0$ plot" is visually checked carefully and determines whether it is suitable for the retrieval of snow depth. Suppose one site shows relatively long and stable $h_0$ values during the entire observation period. In that case, i.e., the "$h_0$ plot" has a relatively "flat" segment on the curve, which indicates that this site is qualified to determine the initial range of the snow-free surface reflector height. Afterward, a range of $h_0$ is given manually to narrow the good $h_0$ values. The difference of the minimum and maximum value of the range is set to be no more than 0.5 m. The finalized snow-free surface reflector height for each satellite, each quadrant, and each GNSS frequency are respectively determined as the mean value of the good heights of the ten days. In contrast, if one site has no "flat" segment on the "$h_0$ plot", this site is determined as a low-quality site and will not be used for snow depth retrieval. It should be noted that during this processing step, it can only eliminate those sites with poor data quality for snow depth retrieval rather than distinguishing high- and medium-sites. There are no apparent differences for the high- and medium-quality sites regarding the natural environment. Instead, the medium-quality site is defined using two simple rules, i.e., one is the site has good-quality data, but there is no snow for almost all the years. The other is the site's lack of data for most of the years.

Figure 6 shows the defined rule applied to six individual sites with various surroundings, i.e., bumz, bfhr, bgfc, uqwl, qhdl, and qhbm. The top panel of each subfigure shows the environmental conditions around the station on Google Map, with different colors indicating the footprints for elevation angles of 10°, 15°, 20°, 25°, and 30°, respectively. The bottom panel of each subfigure shows the sorted 10-day reflector heights of snow-free surface (i.e., $h_0$). The plots clearly show the differences in the heights for different sites. The first two sites, i.e., "bumz" and "bfhr", show relatively long and stable $h_0$ values for all the GNSS satellites, quadrants, and frequency bands during the entire observation period. It indicates that these sites are flat enough for all the orientations and are ideal for determining the initial range of the snow-free surface reflector height, i.e., 2.5 ~ 2.8 m for "bumz" and 2.8 ~ 3.1 m for "bfhr". Unlike these two sites, the "bgfc" site has relatively stable $h_0$ values only in specific orientation whose natural condition is open and flat. At the same time, it is impossible to derive correct $h_0$ values for "bgfc" in other orientations that have buildings or trees; This phenomenon can be verified from the photo of the site in Figure 5. This site is also good enough to determine the initial range of the snow-free surface reflector height, i.e., 3.6 ~ 4.1 m. On the contrary, the three sites at the bottom of Figure 6, i.e., uqwl,

qhdl, and qhbm, show continuously changed $h_0$ values because of the poorly defined peaks for most Lomb-Scargle periodograms. It indicates that it is unreliable to determine a true $h_0$ due to complex environmental conditions.

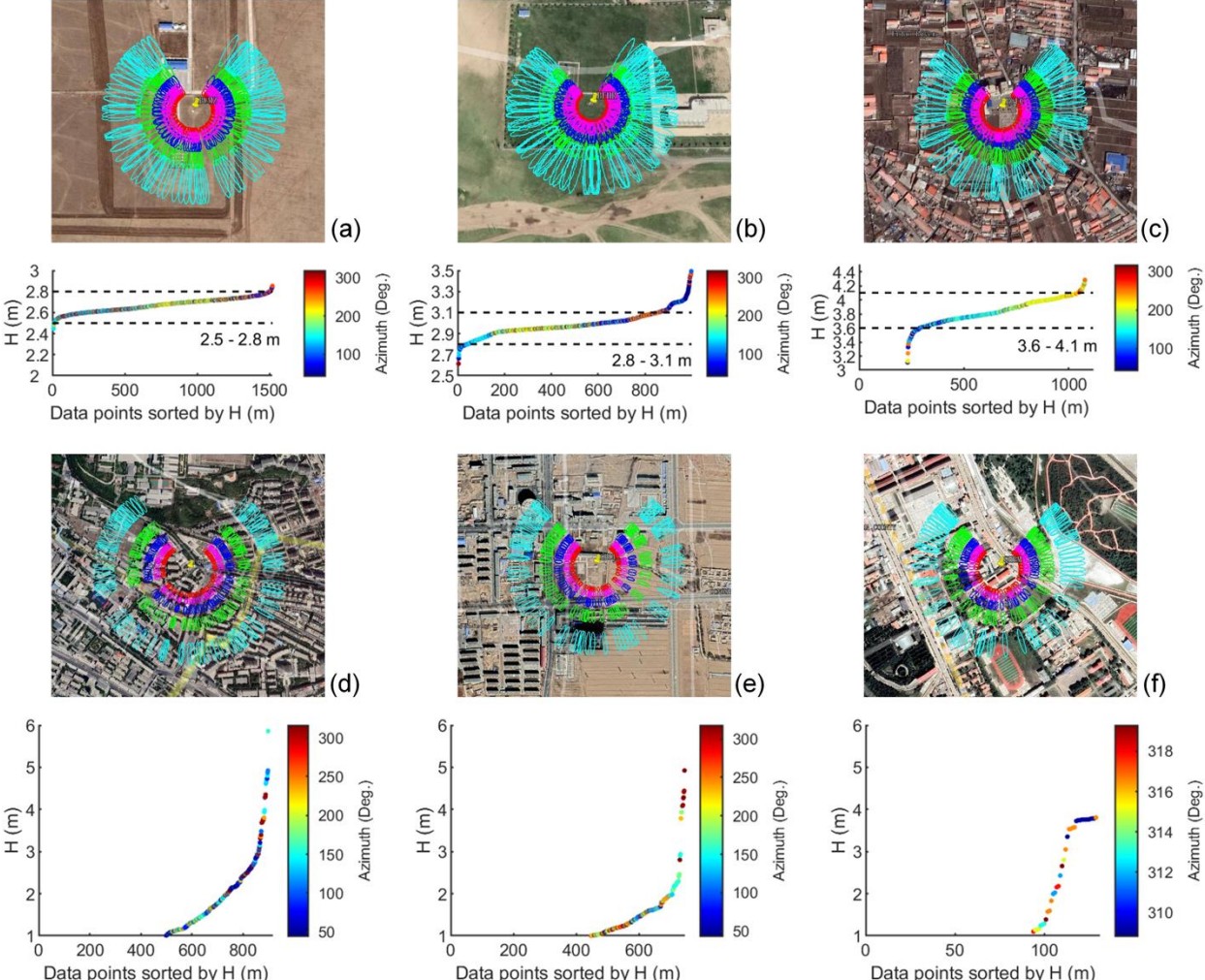

**Figure 6.** Examples show the high/medium-quality sites and the low-quality sites. High/medium-quality sites: (a) bumz, 2017; (b) bfhr, 2019; (c) bgfc, 2019; Low-quality sites: (d) uqwl, 2019; (e) qhdl, 2020; (f) qhbm, 2018. The top image in each subfigure shows the footprint of the observation for elevation angles of 10°, 15°, 20°, 25°, and 30°, respectively. The bottom image in each subfigure shows the distribution of the reflector heights for snow-free surfaces calculated from 10-days of observations using the SNR model. The background of this figure is from Google Earth (https://earth.google.com/web/) © Google Earth 2021.

**3.4 Deriving snow depth of finer resolution**

The default temporal resolution of the snow depth data set is 24-hour. However, some sites have adequate satellite observations that make it possible to produce finer resolution snow depth data. We have two different solutions to produce snow depth of finer temporal resolution. For most sites with only GPS observations, we try to produce 12-hour snow depth if there are no less than five valid observations from 0 ~ 12 UTC or 12 ~ 24 UTC within one specific day. The snow depth value for each 12-hour is defined as the mean of all the observations during this time window. For a few sites with GPS/GLONASS compatible observations, we use the exact processing solutions like the previous GPS-only sites and combine all the observations from the GPS and GLONASS systems to derive finer temporal resolution snow depth. Unlike the previous 12-hour maximum resolution, 2-hour, 3-hour, and 6-hour resolutions can be achieved using compatible observations.

**3.5 Quality control of the snow depth data set**

Several postprocessing steps are executed to accomplish the quality control of the raw snow depth data set. This section gives detailed information on these steps as follows:

(1) Moving average filtering

For each site, as shown in Figure 7, the raw snow depth values over a snow season, i.e., from October 1st this year to April 30th the following year, are gathered together. The moving average algorithm is executed to filter out the snow depth outliers probably due to the incorrect recognition of the peak frequencies on the Lomb-Scargle spectrums. This moving average method is a traditional way to reject outliers (Wang et al., 2020; Tabibi et al., 2017a; Nievinski and Larson, 2014a). Snow depth values out of the 95% confidence interval are smoothed over a sliding window across neighboring elements. The length of the moving window is set to be 12-hour in this study. In the finalized GSnow-CHINA data set, we also provide the original data set without filtering to allow users to check the initial form of the data. The following analyses in Sections 4 and 5 are based on the filtered data.

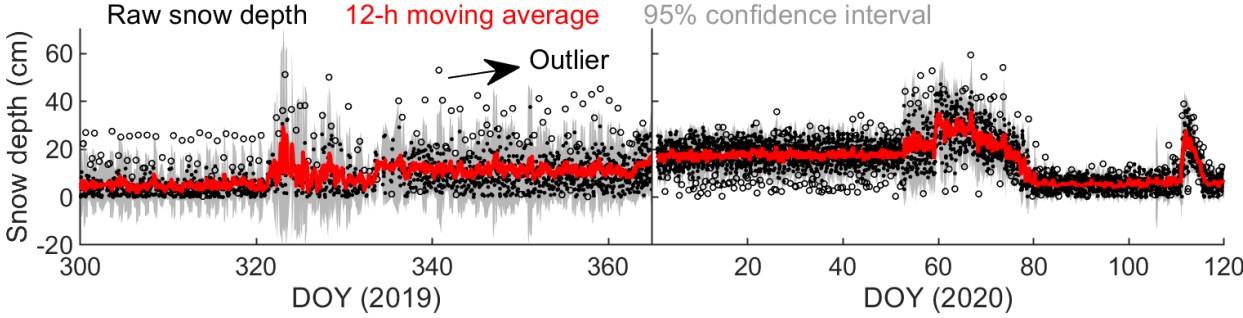

**Figure 7.** Examples showing the moving-average filtering of the snow depth results over one snow season. The site presented in this figure is "bfqe" which is a CMA site. DOY: day of year.

    (2)  Modifying the system errors caused by the penetration depth of soil

The penetration depth of the GNSS signal through bare soil ($h_p$) directly influences the determination of the reflector height of the snow-free surface. The $h_p$ is dependent on the soil permittivity and the GNSS wavelength. The soil permittivity is related to soil moisture and soil components. Figure 8 (a) shows the relationship between penetration depth of GPS L1 band and soil moisture/soil components calculated using parameters provided in (Hallikainen et al., 1985). The penetration depth is deeper than 10 cm when soil is very dry (i.e., volumetric soil moisture (VSM) < 0.1 $cm^3.cm^{-3}$). The penetration depth is around or shallower than 5 cm under normal soil moisture conditions. In this study, the soil components data for each site, i.e., the percentages of sand and clay, are approximatively derived from the China Soil Science Database (http://vdb3.soil.csdb.cn/) by the soil attributes of the specific city and province that the site is located in. The average VSM of each site is calculated as the multiple-year mean value of the SMAP VSM. The penetration depths of each site for GPS L1/L2, GLONASS B1/B2, and BDS B1/B2/B3 are subsequently calculated using the prepared soil components and VSM parameters. Figure 8 (b) shows the number of GNSS sites categorized by the soil penetration depths ($h_p$). The majority has a shallow penetration depth of 4~8 cm, with only a few having 10 cm or deeper. The $h_0$ is modified as $(h_0 - h_p) + C$ for the final production of the snow depth data set. C is an empirical constant set as 3 cm in this study to represent the offset of the complicated land surface conditions.

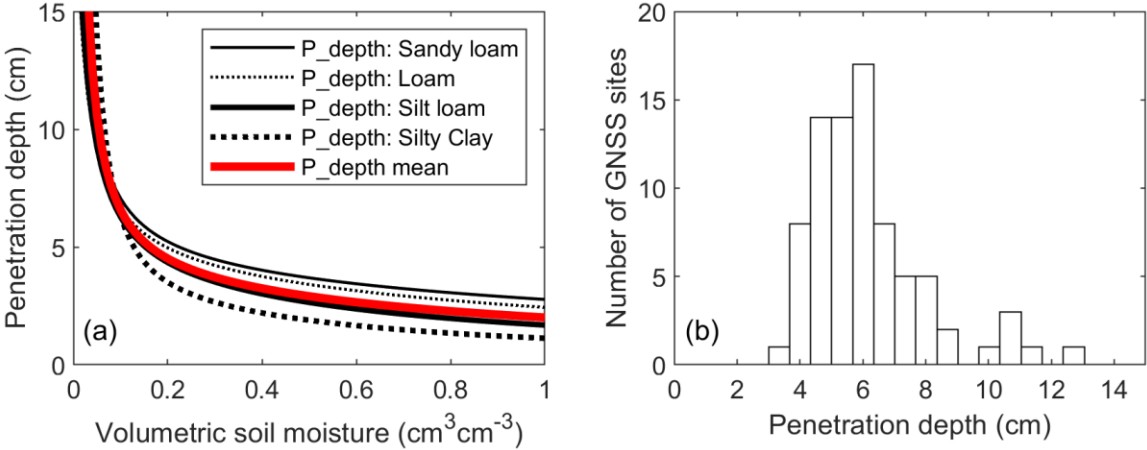


**Figure 8**. (a) The penetration depth of GNSS signals over the soil layer, taking GPS L1 band (wavelength = 19 cm) as an example. The red line indicates the mean penetration depth for various soil types; (b) Statistics of the number of GNSS sites categorized by the soil penetration depths (also taking GPS L1 band as an example).

380       (3)  Eliminating the vegetation effects

For densely vegetated surfaces, particularly in Autumn, vegetation height is usually calculated as "fake snow depth" due to similar responses on the Lomb-Scargle spectrum. However, it is difficult to identify whether it is vegetation or snow. As for northern China, this phenomenon occurs mainly in October and early November. In this study, for each site from October 1st to November 15th, if there are snow depth records from the GNSS data,

we use the NDVI from MODIS data and the historical weather report to determine whether it is actual snow or not. After this round of checking, to ensure the reliability of the snow depth, for 15 sites that probably have "fake snow depth" records, DOYs 270 ~ 300 are masked out from the data set.

      (4)  Quality flags

The number of GNSS satellites used for this calculation is used as a quality flag for each snow depth data

record. In this study, we set the threshold to be 5 to preserve as much data as possible. According to this quality flag, the users can decide whether to use a snow depth data record with a low number of observations. For each snow depth data record, the STE of the snow depths for different satellite tracks is treated as another qualifying flag. The users can also decide their own rules to filter the data according to this quality flag. The 8-day MODIS NDVI is also included as a quality flag in the data set to show the vegetation conditions of the site initially. The

8-day values are combinations of the MODIS MOD13Q1 and MYD13Q1 products. The NDVI flag can provide

supplementary information for the users to identify the possible error due to vegetation. However, due to the coarse resolution of MODIS data, it is not possible to use this flag to represent the actual vegetation cover around the GNSS station.

**3.6 Error indicators used in this study**

The Root Mean Square Difference (RMSD), normalized RMSD (nRMSD), STE, and normalized STE (nSTE) are four error indicators used in this study. The RMSD of two data ($X$ and $Y$) are given by $RMSD = \sqrt{\sum(X_i - Y_i)^2 / N}$, where $N$ is the number of elements in the sample. The nRMSD is given by $nRMSD = RMSD/(\text{mean}(X))$. The STE of one data ($Z$) is given by $STE = \sigma_Z/\sqrt{N_Z}$, where $\sigma_Z$ is the standard deviation of the data $Z$ and $N_Z$ is the number of elements in $Z$. The nSTE is given by $nSTE = STE/\bar{Z}$, where $\bar{Z}$ is the

mean of the sample.

**4    Validation of the data quality**

**4.1 Intra-comparisons of GNSS snow depth results**

The intra-comparisons of the snow depths are executed from three aspects, i.e., comparison of different GNSS constellations, frequency bands, and receivers. If we compare one of the three factors, we should prevent

the other two and other random errors from cross-influence. In other words, we should ensure a snow depth value is "accurate" under the defined condition. Therefore, in this section, we use a rigorous threshold of STE = 1 cm to filter out the outliers. We show the correlation coefficient (r), RMSD, and nRMSD values for each comparison. It should be noted that the nRMSD (snow depth > 5 cm) is significantly lower than the nRMSD (all), which is because the reference value (i.e., the mean snow depth) was used to normalize the RMSD. A large portion of

snow depths in the study area is lower than 5 cm, yielding a lower mean value when involving all the data than only using the > 5 cm data. The same principle applies to the following Figures 9, 10, and 11. Nevertheless, the metrics only represent the comparison during the intermediate process of the data set production. Users can define their own rules to use the data according to the quality flags in the published data set.

Figure 9 (a) (b) shows correlations of the snow depths between GPS and GLONASS for 24-hour and 12-

hour respectively, using data from the four GPS/GLONASS compatible sites. Both show good agreement, with the correlation coefficient r = 0.98 and with RMSD = 1.01 cm for the 24-hour result and RMSD = 0.97 cm for the 12-hour results. Figure 9 also shows the RMSD and nRMSD values of snow depths greater than 5 cm, which

is within the accuracy of the current GNSS-IR technology. The RMSD (nRMSD) of the 24-hour and 12-hour results are respectively 1.65 cm (0.11) and 1.51 cm (0.10). The BDS results are not used for comparison due to the limited number of observations.

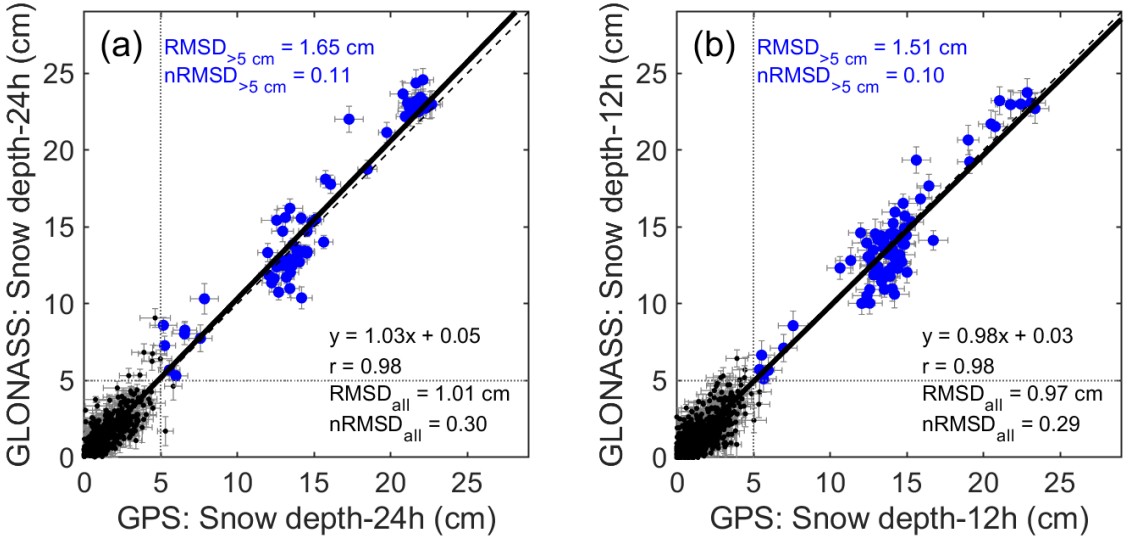

**Figure 9.** Correlations of 24 h/12 h snow depths from GPS and GLONASS observations. (a) 24 h; (b) 12 h. The error bar of each point is the standard error (STE) of the snow depths for all the available tracks of this point. Four available sites, i.e., hltl, hlhl, bfqe, and bttl, during the GPS/GLONASS overlapped periods (i.e., the year 2014 and 2015) are used to plot this figure. For each point in the figure, the number of valid observations is more than five. To prevent other possible effects besides the GNSS constellation, the STE of snow depths is less than 1 cm (90% for the 24 h data and 76% for the 12 h data). Blue points are with the retrieved GPS and GLONASS snow depths greater than 5 cm. RMSD: Root Mean Square Difference; nRMSD: normalized RMSD.

Figure 10 (a1, a2) and (b1, b2) shows correlations of the snow depths between GPS L1 and L2 and between GLONASS L1 and L2, respectively, using data from the same four GPS/GLONASS compatible sites as in Figure 9. The results from different frequency bands show good consistency with each other, with r = 0.94 (RMSD = 1.64 cm) for GPS, and r = 0.99 (RMSD = 1.28 cm) for GLONASS (Figure 10 (a1) and (b1)). The RMSD (nRMSD) values of snow depths greater than 5 cm are 2.68 cm (0.22) for GPS and 1.86 cm (0.10) for GLONASS. It should be noted that a small part of the difference between L1 and L2 is due to the antenna phase centers not being in the same place. The initial bias occurs on the raw L1 and L2 reflector heights. However, the final bias becomes

negligible because, during snow depth calculation, the reflector height value of bare soil is subtracted. The BDS results still are not used for comparison due to the limited number of observations.

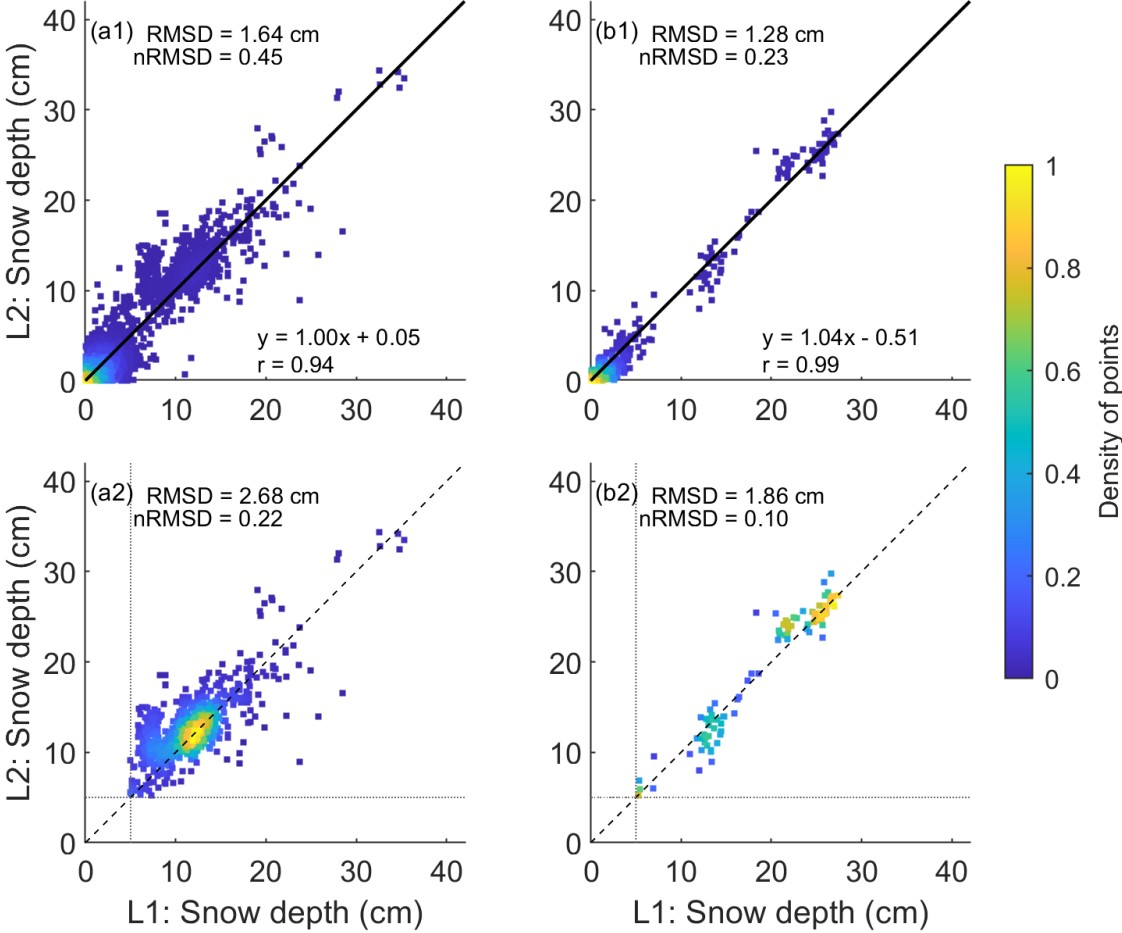

**Figure 10.** Correlations of snow depth from different GNSS frequencies. (a1) GPS L1 vs. GPS L2; (b1) GLONASS L1 vs. GLONASS L2. The color bar represents the density of points; (a2) Same as (a1) but with snow depths greater than 5 cm; (b2) Same as (b1) but with snow depths greater than 5 cm. Fifty-one high-quality GPS sites of CMA and four GPS/GLONASS compatible sites are respectively used to plot (a1, a2) and (b1, b2). For each point in the figure, the number of valid observations is more than five. To prevent other possible effects besides the GNSS frequency,

the STE of each snow depth is less than 1 cm in all the subfigures (61% for the GPS data and 70% for the GLONASS data). RMSD: Root Mean Square Difference; nRMSD: normalized RMSD.

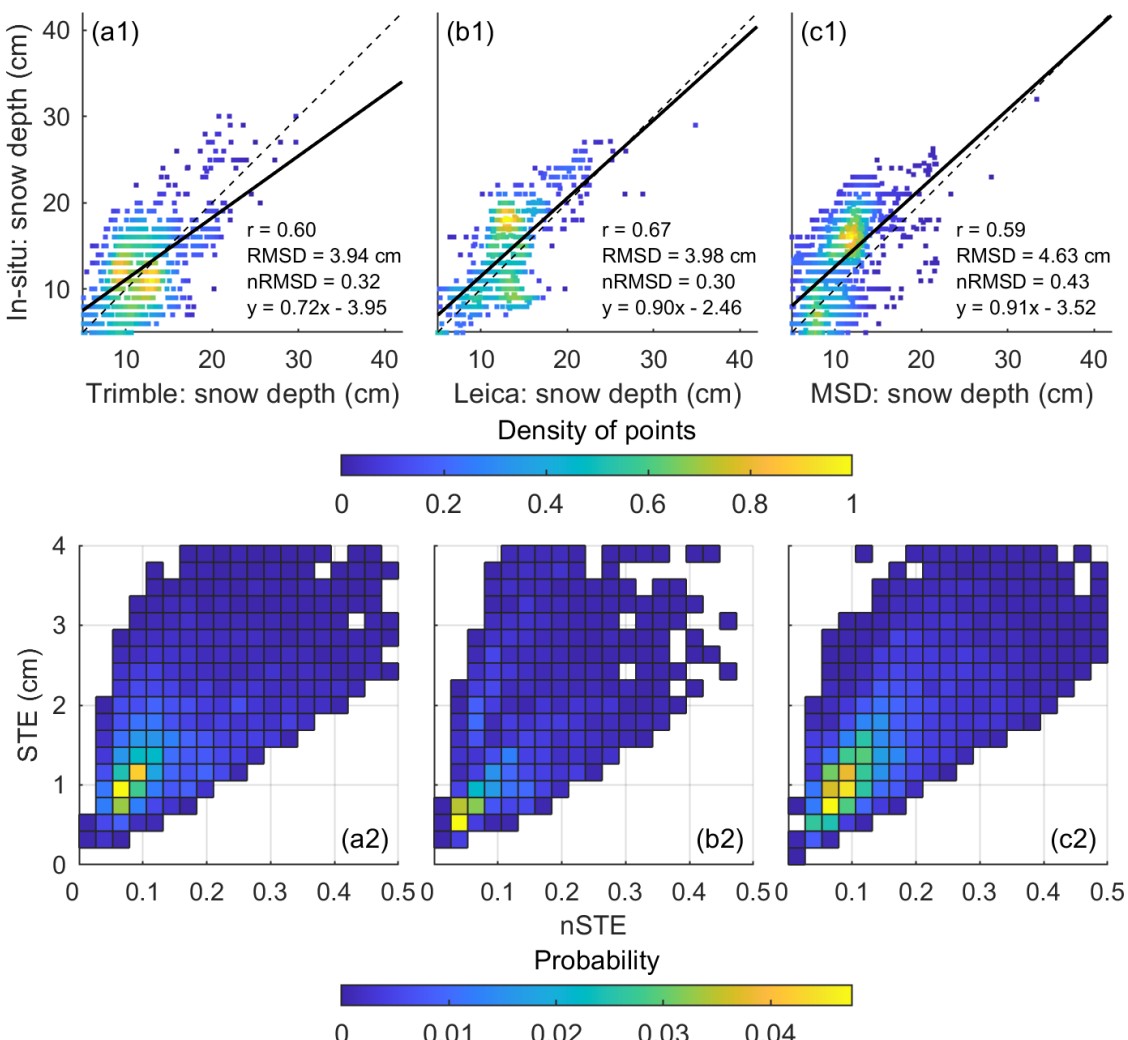

**Figure 11.** Comparisons of the GNSS-derived snow depth and the in-situ measurements from different types of

GNSS receivers: (a1) Trimble; (b1) Leica; (c1) Minshida (MSD), and the histogram of the standard error (STE) and

nSTE of snow depths for different types of GNSS receivers: (a2) Trimble; (b2) Leica; (c2) MSD. The number of

sites representing Trimble, Leica, and MSD is 20, 5, and 24. The GNSS snow depths values are greater than 5 cm in

this figure. To prevent other possible effects besides the receiver type, the STE of snow depths is less than 1 cm

(63% of the entire data) in (a1), (b1), and (c1). RMSD: Root Mean Square Difference; nRMSD: normalized RMSD.


The CMA and CEA sites are set up with various brands of GNSS receivers. Most of these receivers are from

three brands, i.e., Trimble, Leica, and MinShiDa (MSD). Taking these three brands as examples, in order to

evaluate the snow depth results from these three brands, Figure 11 (a1), (b1), & (c1) respectively show the

differences of the snow depths derived from the three brands, taking the in-situ measurements as benchmarks.

The results from the three brands show good consistency with r = 0.60, 0.67, & 0.59 and RMSD = 3.94 cm, 3.98

cm, & 4.63 cm, respectively. Figure 11 (a2), (b2), & (c3) further show the histogram of the STEs and nSTEs of

the snow depths from the three brands, and good consistency is also shown in these subfigures. The maximum

of the statistical STE (nSTE) for Trimble, Leica, and MSD is respectively around 1 cm (0.07), 0.6 cm (0.04), and

1 cm (0.07). Due to the inconsistent footprint between the GNSS and in-situ measurements, the error metrics

presented in Figure 11 are for reference only and do not represent factual accuracies.

From the comprehensive intra-comparisons shown in Figures 9 ~ 11, we conclude that the snow depths

derived from different GNSS constellations, frequency bands, and receivers have overall good agreement. The

average values of the metrics shown in Figures 9 ~ 11 are summarized as follows: mean r = 0.98, mean RMSD

= 0.99 cm, and mean nRMSD (snow depth > 5 cm) = 0.11 for different GNSS constellations, mean r = 0.97,

mean RMSD = 1.46 cm, and mean nRMSD (snow depth > 5 cm) =0.16 for different frequency bands, and mean

r = 0.62 for different GNSS receivers. Therefore, it is feasible to combine all these results to produce the snow

depth data set in this study.

**4.2 Comparison with in-situ measurements and the PMW products**

The GNSS snow depth data set, the PMW data set, and the in-situ measurements are not consistent in terms

of the spatial footprint. The GNSS and in-situ data have a closer footprint than the 25-km PMW data. The

footprint of GNSS is approximately ~ 30 m x 30 m, as illustrated in the following Figure 17. Due to the

discrepancy in footprint, it is impractical to give factual accuracies when comparing these three data sets. Instead,

we present the performance of the three data sets at daily scale, multi-year scale, and interannual variabilities.

The RMSD and nRMSD values presented in Figure 13 and Figure 14 are for reference only and do not represent

factual accuracies.

Figure 12 shows an example of the comparisons of daily snow depth derived from GNSS, in-situ, and PMW.

The data used in this figure is from 16 GNSS sites in 2016-2022, with the least missing daily snow depth values.

The comparison period is from 2016 to 2022 due to the data discontinuity in other periods. The three data sets

have similar variation trends but with apparent differences in absolute snow depth values. The GNSS-derived

snow depths are closer to the in-situ values than the PMW for most sites because GNSS and in-situ have a closer

footprint. However, for some sites (e.g., Site "jldg" in Figure 12), the in-situ measurements are much higher than

the GNSS and PMW, which needs further in-depth analysis. Figure 12 presents all the GNSS snow depth values of the 16 GNSS sites, regardless of its quality, to give a comprehensive illustration of the data. It is recommended that the users define their own rules to determine whether to use those snow depth values with low numbers of

GNSS tracks or high STEs.

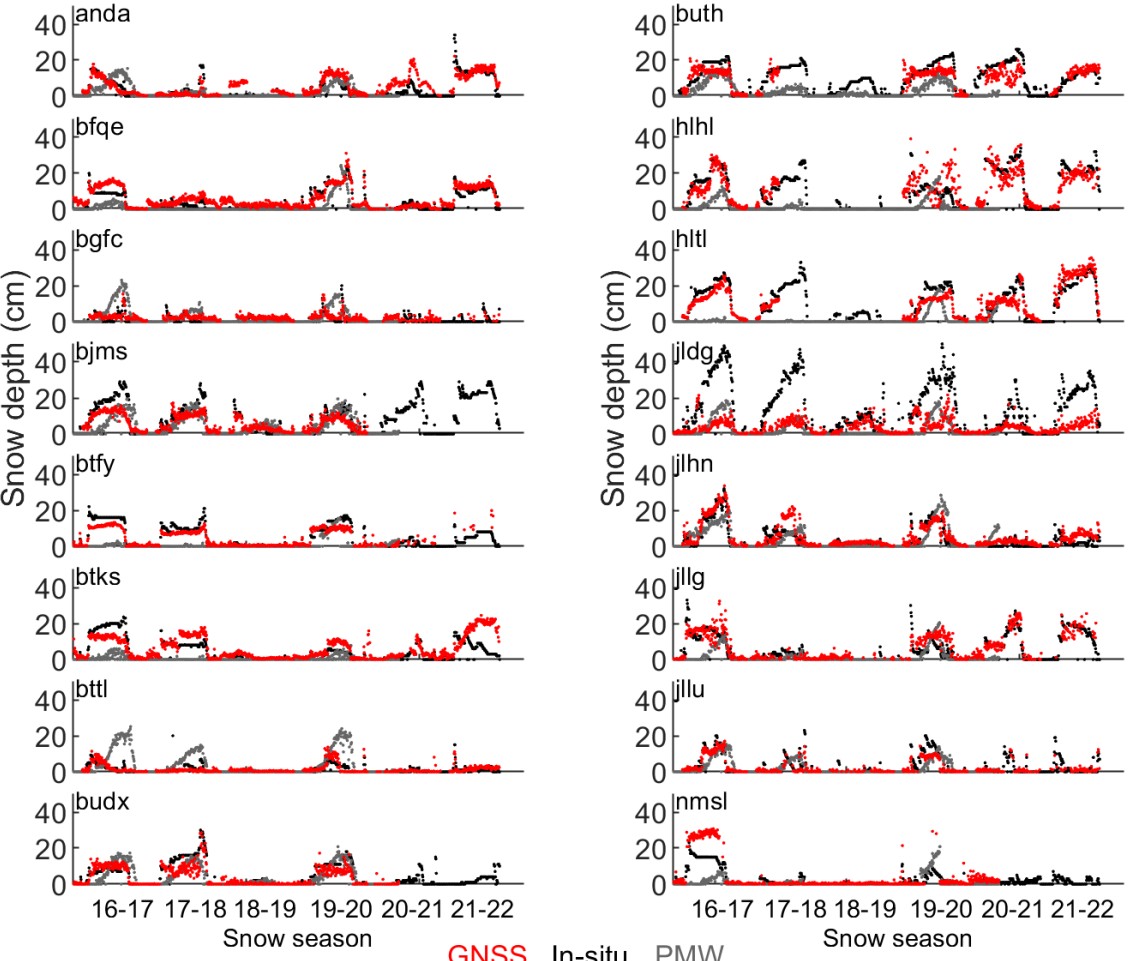

**Figure 12.** Comparisons of daily snow depth derived from GNSS, in-situ, and PMW. The data used in this figure is from 16 GNSS sites in 2016-2022, with the least missing daily snow depth values.

Figure 13 shows an example of the comparisons of daily mean snow depth derived from GNSS, in-situ, and PMW. The data used in this figure is from 17 GNSS sites with the most extended temporal coverage (i.e., from 2013 to 2022). As expected, the GNSS and in-situ data have similar performance compared to the PMW data, with RMSD = 2.37 cm & nRMSD = 0.23 for GNSS vs. in-situ, and RMSD = 3.55 cm & nRMSD = 0.35 for

GNSS vs. PMW. In addition, the peak of the PMW snow trend for each snow season is later in the season, which

is due to the change of snow grain size (Dai et al., 2012).

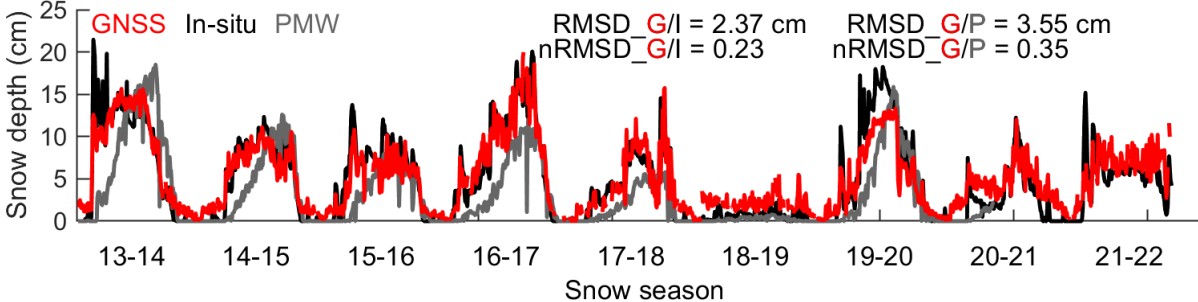

**Figure 13.** Comparisons of daily mean snow depth derived from GNSS, in-situ, and PMW for17 GNSS sites with

the most extended temporal coverage (i.e., from 2013 to 2022). RMSD: Root Mean Square Difference; nRMSD:

normalized RMSD.


The annual mean and maximum snow depths are significant indicators that can reflect the overall data

quality and the variation trend over multiple years. Sixteen sites with the least missing daily snow depth values

(the same as data used in Figure 12) are used to compare the multi-year averages of the annual maximum/mean

snow depth derived from GNSS, in-situ, and PMW. Coincidentally, all these sixteen sites are located in the NCM

region, making it possible further to analyze the interannual variability of the multi-year maximum or mean snow

depth. Figure 14 shows a site-by-site comparison of the five-year average of the annual maximum /mean snow

depth derived from GNSS, in-situ, and PMW, respectively. Figure 14 (a1) & (b1) respectively show the spatial

distribution of 16 sites marked by their corresponding values of the average of the annual (a1) maximum and (b1)

mean snow depth. The snow depth values are classified into five categories to show consistency and discrepancy

better. It shows high consistency for the three data sets in general but with discrepancies for some sites. Figure

14 (a2) & (b2) respectively show the site-by-site comparison of the average of the annual (a2) maximum and (b2)

mean snow depth.

The maximum values are consistent for the three data sets without regard to the in-situ data having one

outlier at Site jldg. This data point is an outlier because the historical weather reports showed no significant

snowfall events before or after these dates. This result is a reminder that operational laser measurements of snow

depth are not always reliable. For the mean values shown in (b2), the GNSS and in-situ have a better agreement

than the PMW because of the significant difference in their spatial footprint. Most sites are located in the region with evergreen coniferous forest, which prevents the PMW data from acquiring reliable snow depth values due to its wider observation extent of 25 km. Figure 14 (a3) & (b3) further show the correlation between the GNSS

and in-situ or PMW. Accordingly, higher consistencies are achieved from GNSS vs. in-situ than GNSS vs. PMW, with r = 0.75 (RMSD = 4.08 cm) vs. r = 0.57 (RMSD = 6.10 cm) for the maximum and r = 0.90 (RMSD = 1.22 cm) vs. r = 0.75 (RMSD = 3.59 cm) for the mean. The outliers are not involved during the correlations.

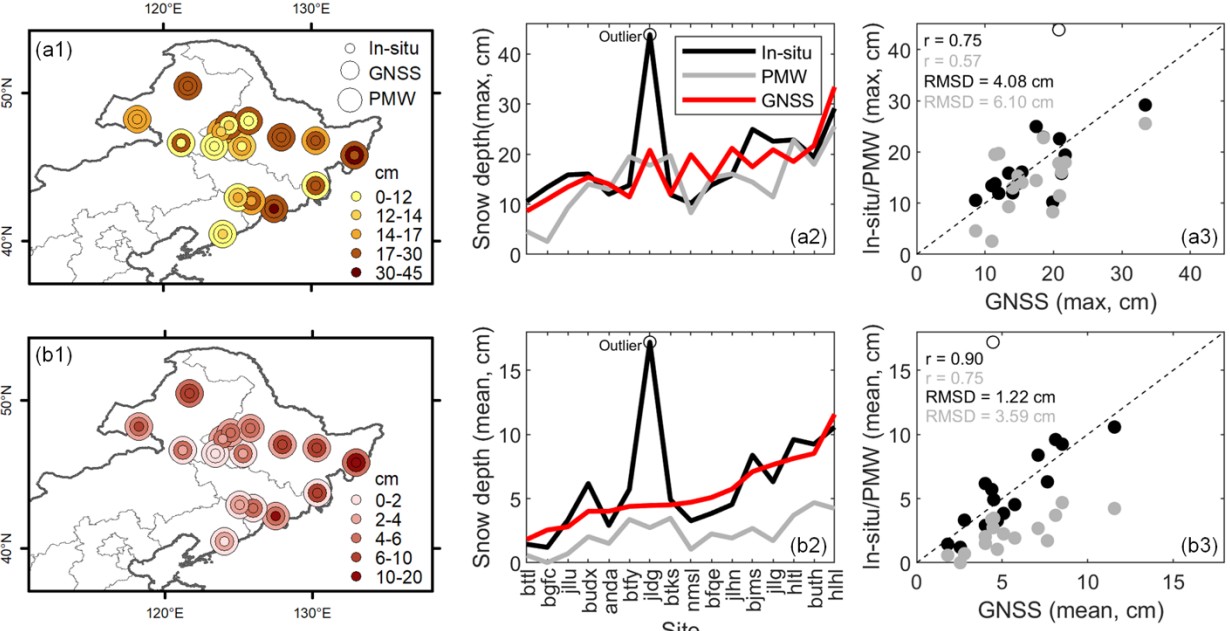

**Figure 14.** Site-by-site comparison of the five-year average of the annual maximum /mean snow depth derived from

GNSS, in-situ, and PMW, respectively. (a1) The spatial distribution of the sites marked by their corresponding values of the five-year average of the annual maximum snow depth; (b1) Same as (a1) but the annual mean; (a2) The site-by-site comparison of the five-year average of the annual maximum snow depth; (b2) Same as (a2) but the annual mean; (a3) The correlation between the GNSS and in-situ/PMW for the five-year average of the annual maximum; (b3) Same as (a3) but the annual mean. Sixteen sites with the least missing daily snow depth values from

2016 to 2022 are used to draw this figure. The site names are shown in (b2). RMSD: Root Mean Square Difference.

The interannual variability of the multi-year average of the annual maximum (mean) snow depth using the same data in Figure 14 is further shown in Figure 15. The snow depth values in this figure are the mean values of all 16 sites. The maximum and mean achieve consistent interannual variabilities for all the three data sets, with the

absolute maximums of the PMW being relatively higher than the other two. This result generally indicates that

the GNSS data set in this study can be used as a new data source to monitor the interannual variability of snow

depth.

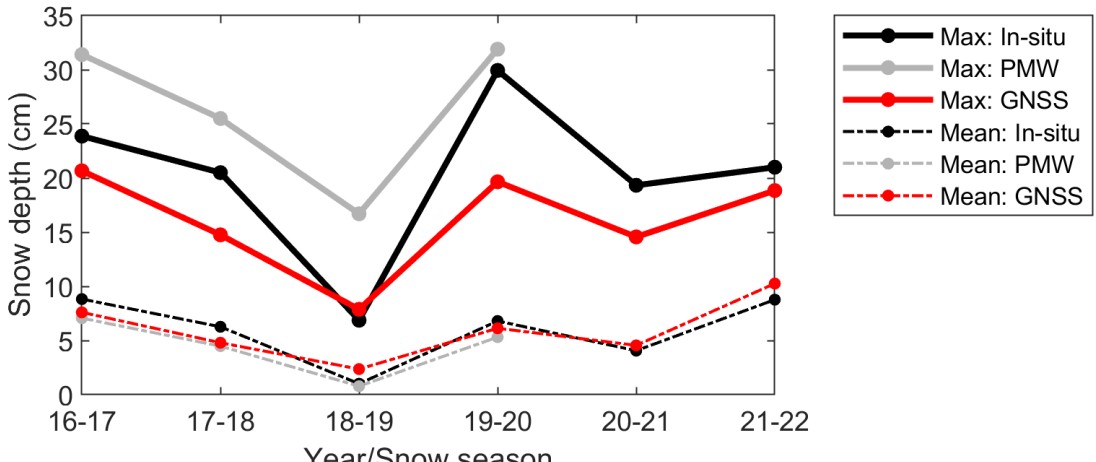

**Figure 15.** Interannual variability of the multi-year average of the annual maximum (mean) snow depth derived from

GNSS, in-situ, and PMW. Sixteen sites with the least missing daily snow depth values from 2016 to 2022 are used to

draw this figure. The site names are shown in Figure 14 (b2). The PMW data were available only for the period

2016-2020.

**4.3 Reflection on extreme snow event**

Real-time and accurate monitoring of extreme snow events is of vital practical value. To test if this new

GNSS data set can provide supportive information for this application, we use the extreme snow event that

happened on February 21 ~ 22 in the year 2015 to analyze the performance of the GNSS, in-situ, and PMW data

sets. The event is selected because we have overlapped GNSS data from two GPS/GLONASS compatible sites,

i.e., bfqe and bttl, which can provide finer resolution snow depth observations. Figure 16 (a) shows the daily

snow depth variations before and after the snow event. As expected, the GNSS and in-situ data have similar

responses to the event, while the PMW data has a weak response. As indicated previously, these two sites are

located in the evergreen coniferous forest region, which prevents the PMW data from acquiring reliable snow

depth values due to its much larger footprint of 25 km. Figure 16 (b) further shows the response of the 6-hour

GNSS snow depth data during the week of the event. It captures the evolution of the event in a more detailed

way from DOY 51 than that of the other two data sets. However, due to the lack of reference data at the same

rate, it is impossible to evaluate the quality of the 6-hour GNSS data set. There are several discontinuities in the

GNSS-derived snow depth (i.e., sharp decrease or increase) that are typically not seen in snowstorm data. The

common feature of these abnormal values is they all have high STEs (as shown in the bottom panel of Figure 16

(b)). As shown in the top panel of Figure 16 (b), it is possibly due to the relatively low number of tracks used for

producing the data set. The 2-hour and 3-hour data are not shown in the figure due to severe data missing for

some periods. Regardless of the limitations mentioned above, the GNSS data provides the potential to increase

the monitoring frequency of extreme weather in a cheap and effective way in the future, even with a higher

resolution of 6-hour or better, particularly for those sites that have compatible observations from more GNSS

satellite systems such as GPS, GLONASS, BDS, and Galileo.

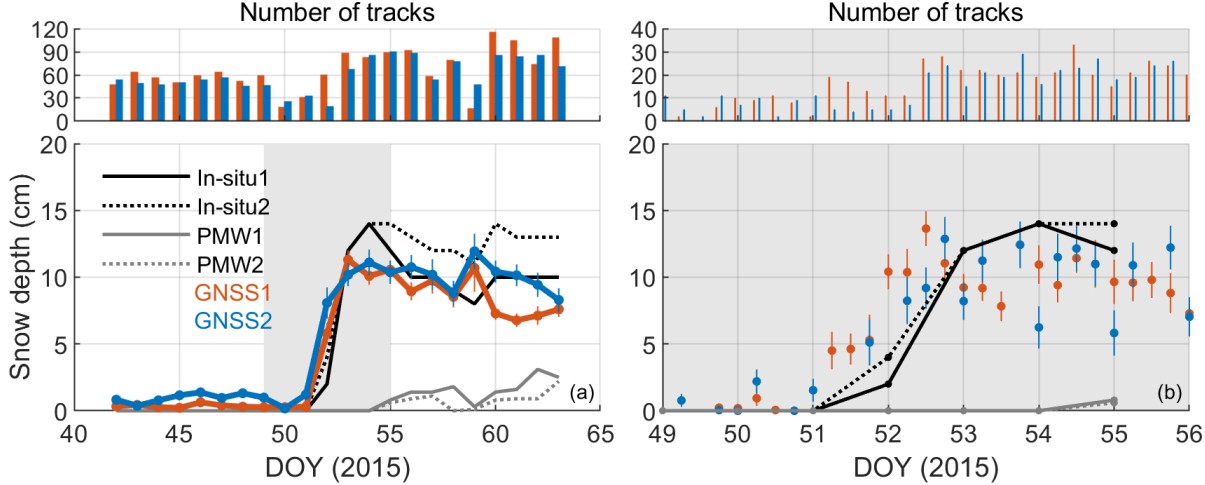


**Figure 16.** Performance of the GNSS snow depth on a snow event. (a) Daily data; (b) Two-hour data. Two

GPS/GLONASS compatible sites, i.e., bfqe (in red) and bttl (in blue), are used to draw this figure. The error bar of

each point in the figure is the standard error (STE) of the snow depths for all the available tracks of this point.

**5   Data set descriptions**

The GSnow-CHINA Version 1.0 data set is developed using observations from the two GNSS networks

constructed by the CMA and CEA. The data set is available at National Tibetan Plateau/Third Pole Environment

Data Center via https://doi.org/10.11888/Cryos.tpdc.271839 (Wan et al. 2021). It is called Version 1.0 because

we produce the data set using historical observations till the year 2022, and there is room for improvement of the

algorithm (e.g., how to properly consider the effects of vegetation and terrain). We will continue to maintain and

update the algorithm and the data set as more years of data become available in the future. The data set includes

snow depth of 24-hour, 12-hour, and 6/3/2-hour temporal resolutions if possible, for 80 sites from 2013 ~ 2022

over northern China (25° ~ 55° N, 70° ~ 140° E). The sites over southern China are not included because there

is most probably no snow in that region. The high and medium sites are all preserved in the data set with multiple

quality flags for users to apply the data.

There are two folders in the data set, i.e., the SITE_INFO and the SNOW_DEPTH. The SITE_INFO folder

includes the general information of the 80 GNSS sites, with four separate sheets in one .XLS file corresponding

to CMA high-quality, CMA medium-quality, CEA high-quality, and CEA medium-quality, respectively. The

items in the file are listed as SITE_NAME, LAT (latitude), LON (longitude), ALT (altitude), RECEIVER_TYPE,

GNSS_TYPE, ANTENNA_HEIGHT (in meter), and MEAN_VSM (Volumetric soil moisture in $cm^3cm^{-3}$; Mean

value derived using SMAP soil moisture data of 2015-2020). The SNOW_DEPTH folder includes the snow

depth values for all available sites. The folder is structured by ~/site/. For example, ~/hltl/stores the snow depth

data of the Site hltl. There are four sub-folders in the folder of each site, i.e., raw0, filtered0, raw, and filtered.

The "raw0" and "filtered0" folders store raw data and raw-but-filtered data for individual

satellite/quadrant/frequency/time. The "raw" and "filtered" folders store 24-hour/12-hour data produced using

raw data in the corresponding "raw0" and "filtered0" folders. The file names including *_24h.csv, *_12h.csv,

and *_02h.csv represent the 24-hour, 12-hour, and 2-hour resolution data. Each CSV file gathers this specific

snow season (e.g., the 2019 file stores values from October 1, 2019, to April 30, 2020). We recommend using

the snow depth data in the "filtered" folder for validation/application purposes while using the snow depth data

in the "raw" folder for algorithm testing purposes.

Three quality flags are included in each snow depth file, i.e., the STE, NUM_OF_PRNS, and NDVI,

standing for the STE of snow estimations, the number of GNSS sites, and the MODIS NDVI value. These flags

should be used to filter the data to balance the data volume and the snow depth accuracy. In addition, we do not

recommend using the snow depth values of less than 5 cm in the data set, which is beyond the accuracy of the

current GNSS-IR technology.

Figure 17 shows an example of the snow sensing footprint for a specific satellite track. For a 3-meter antenna height under regular 10° ~ 30° elevation angles, the footprint of a specific satellite track is defined as ellipses characterized by the First Fresnel Zone (Larson and Nievinski, 2013), with the maximum length of  ~ 30 m for one direction. The GNSS footprint can be recognized as a ~ 30 m x 30 m circle for all orientations. This footprint is between the point-scale of the in-situ measurements and the course 25-km resolution of PMW, which makes it an effective supplement data source for research, validation, and application purposes.

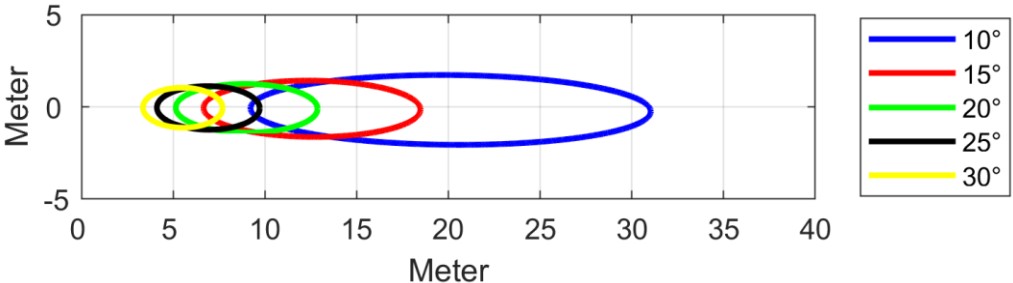

**Figure 17.** The footprint of the GNSS snow depth observation for a specific satellite track with different satellite elevation angles.

## 6    Extended analysis of the data set and method

Although this study releases a data set using the current GNSS sites, which are suitable for snow depth retrieval. Those sites that are not suitable for this purpose still deserve an extended analysis to promote this research domain's development further. Also, although the method to retrieve snow depth used in this data set is determined as the SNR model due to data availability, it deserves an extended discussion of the selection of the method for interested readers who dedicates to developing their own data set. The following Sections 6.1 and 6.2 give an extended analysis of the two issues mentioned above.

### 6.1 Factors that affect the site quality for snow depth retrieval

(1) Natural surroundings. The natural environment within the footprint of the observations is the most significant factor that determines whether a specific GNSS site is suitable for snow depth retrieval or not. Open and flat ground with no vegetation is the ideal environment to set up a snow site. In other words, terrain and vegetation are the two significant issues that affect snow depth retrieval.

In practical applications, none of the planar surfaces is entirely horizontal. Small ground tilting angles translate into several tens of centimeters of bias due to the large horizontal distances involved (Larson and Nievinski, 2013). Figure 18 shows simulations for a 2-m antenna height with a variety of snow depth levels and positive terrain slopes using the open-source GPS multipath simulator provided by (Nievinski and Larson, 2014b). For slopes of 5° and less, the error in snow depth retrieval is below 10 cm, while for larger slopes (e.g., 8° in the figure), the residual effects are ~ 15 cm and higher. Fortunately, for GPS satellites with repeatable ground tracks, such a topographic bias remains stable over time. It thus could be canceled out when using Eq. (1) to estimate snow depth, most of which is the case in this study. While for GNSS satellites like GLONASS and BDS, whose ground tracks are non-repeatable, the terrain effect should be considered. Some previous studies investigated methods to eliminate the influence of terrain (Zhang et al., 2017; Zhang et al., 2020). We are also developing a new approach to consider the terrain effects, which will be demonstrated in a future study.

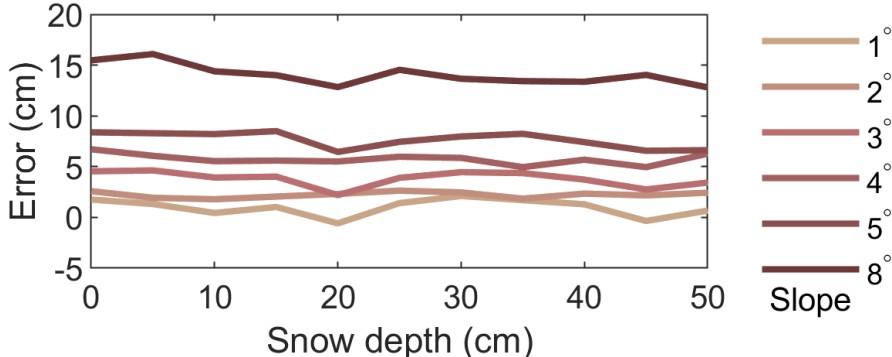

**Figure 18**. Simulations of the effects of terrain slopes on snow depth retrievals for a 2-m antenna height of GPS L1 (wavelength = 19 cm)

Vegetation is another factor that needs to be considered for accurate retrieval of snow depth. Figure 19 shows an example of Site bfxc, which has vegetation effects on snow depth retrieval before DOY 300 for 2015 ~ 2019. The vegetation information is presented by the MODIS 1-km 8-day Normalized Difference Vegetation Index (NDVI) data. The period of the vegetation effects for different years are different, e.g., the years 2016 and 2017 has the most extended period of ~ 30 days from DOY 270 to 300, while the years 2018 and 2019 has only ~ ten days around DOY 270. The effect of vegetation is not strictly consistent with the variation of NDVI, which makes it impossible to build a model to qualify the vegetation effect using NDVI data.

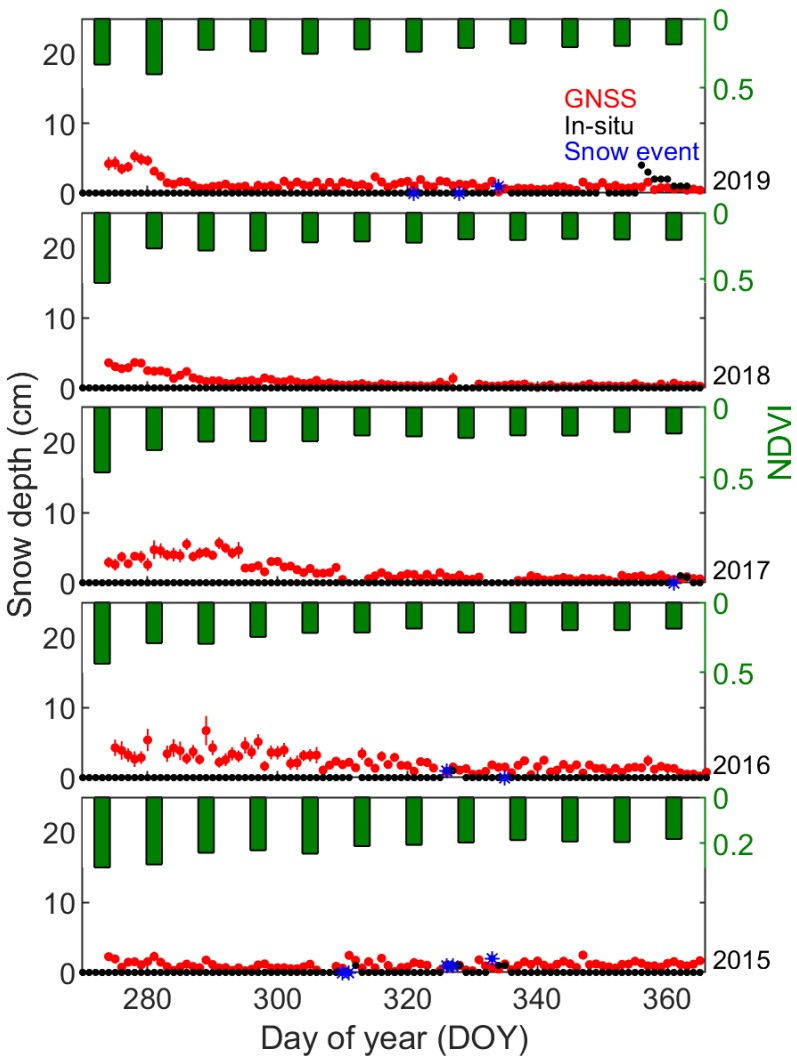

**Figure 19.** Examples showing the vegetation effects on snow depth retrieval. The site presented here is bfxc (2015-2020). The error bar of each point in the figure is the standard error (STE) of the snow depths for all the available tracks of this point.

Figure 20 shows a correlation between the GNSS snow depth and the in-situ measurement colored by NDVI. Note that for those points on the x-axis with in-situ values equal to 0 but have various GNSS snow depth values, the NDVI values are generally higher than other data points. It illustrates that GNSS measures vegetation rather than snow for these data points. A previous study suggested that it is practical to use the amplitude of the GNSS SNR data to retrieve vegetation height for observations of 1-second sampling (Wan et al., 2015). Therefore, for GNSS observations at the sampling intervals, it may be possible to use the SNR amplitude to build a model to

qualify the vegetation effect on snow depth retrieval. However, this is not practical for the CMA or CEA sites used in this study because the sampling interval is 30 seconds for which is impossible to model the SNR data series to derive the amplitude. Future research will consider using other vegetation indicators to identify this issue.

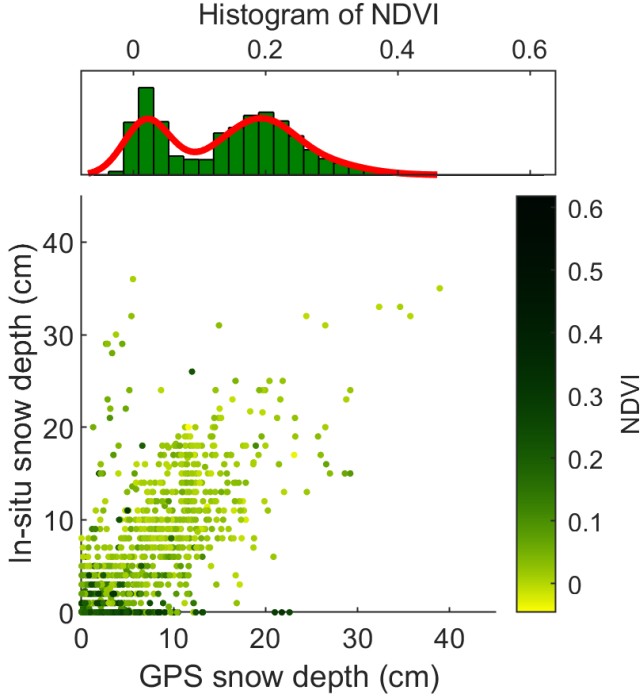

**Figure 20.** Correlation between the GNSS snow depth and the in-situ measurement colored by NDVI. The top panel shows the statistics of the GNSS snow depth when the corresponding in-situ = 0. Three-month data from 74 high- and medium-quality CMA sites are used to draw this figure. For each point in the figure, the number of valid observations is more than five, and the STE of snow depths is less than 2 cm.


(2) Quality of the observation data. The data quality is another critical factor that determines if a site is suitable for snow depth retrieval or not. First, the minimum elevation angle of GNSS satellites should be set to a single number like 5° or 10° to preserve the multipath effect as much as possible because only data with low elevation angles can show the surface reflection. Second, the observables used as inputs for the corresponding snow depth models should be stored in the raw RINEX file. If the stored observables satisfy conditions for multiple models, one can choose the model according to their accuracy or combine to use all the models during the calculation. This issue will be further discussed in the next Section 6.2. Third, the GNSS tracks may miss

data in some epochs during the ascending or descending sequences, although they satisfy the condition of minimum to maximum elevation angles. These data are removed in this study to ensure the accurate acquisition

of the reflector heights. Finally, random errors, e.g., human activities at some point, may exist during the observation.

**6.2 Selection of snow depth models**

Although there are many models to retrieve snow depth, as illustrated in Table 1, considering the availability of the observables and the accuracy of the models, not all models are applicable or optimal in practical application.

Figure 21 shows an overall strategy of model determination for using GNSS data to retrieve snow depth. One should first consider if the SNR observable exists in the RINEX file since the carrier phase and pseudorange are observations that generally exist for positioning. If the observables satisfy all the snow depth models, the optimal model is selected according to the number of frequencies in the RINEX file. If the frequencies received by the receiver are less than 3, the SNR model is the best choice since it is simple and has reliable accuracy (Plan A in

the figure). If the received frequencies are equal to or are greater than 3, the SNR_COM and F3 models can be used (Plan B in the figure). However, one can still use Plan A to replace Plan B in practical applications. If the SNR observable does not exist (Plan C), the F3 model is preferred when the number of CP is greater than 3, while the L4 or the F2C model is selected when the number of CP is less than 3. Nevertheless, the effects of the ionosphere delay on the L4 and F2C models are difficult to remove, which leads to the relatively low accuracy

of these two models (Liu et al., 2022).

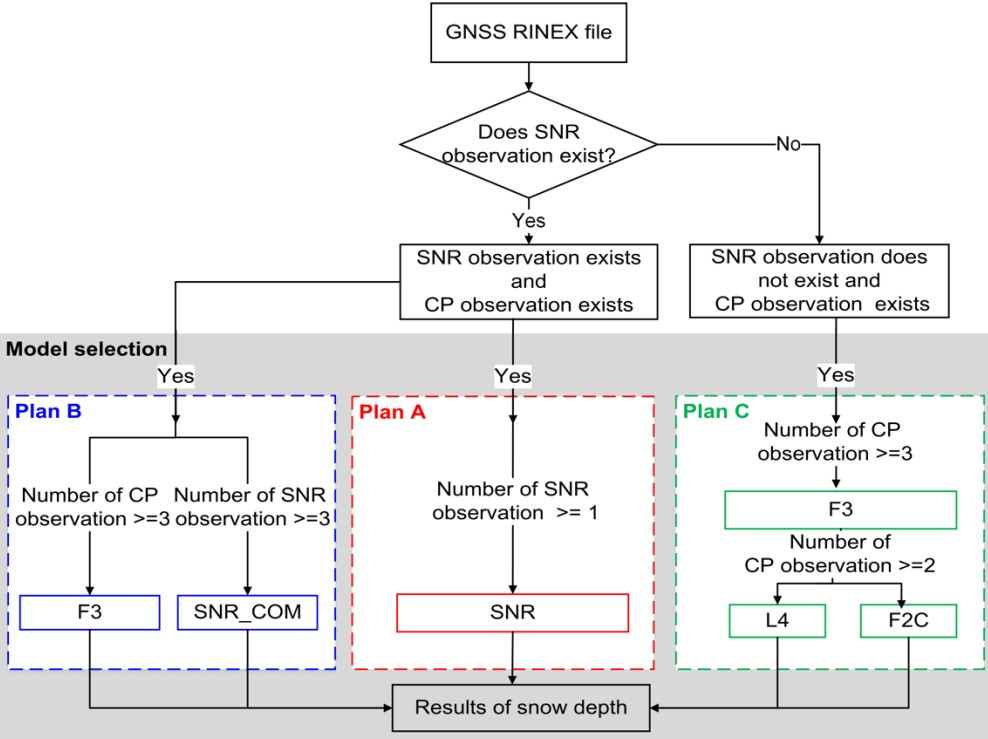

**Figure 21.** The strategy of model selection for using GNSS data to retrieve snow depth. CP: carrier phase. Different solutions are represented as Plan A, B, and C.


## 7 Data availability

The GSnow-CHINA v1.0 data set is archived and available at National Tibetan Plateau/Third Pole Environment Data Center (Li et al., 2020; Pan et al., 2021) via https://doi.org/10.11888/Cryos.tpdc.271839 (Wan et al., 2021).

## 8 Conclusions

This study proposes a comprehensive framework using raw data of the complex GNSS station networks to automatically retrieve snow depth and control its quality. Based on this, this study further produces a long-term snow depth data set over northern China (i.e., GSnow-CHINA v1.0, 12h/24h, 2013-2022) using the proposed framework and historical data from 80 stations.

The data set has high internal consistency with regards to different GNSS constellations (mean r = 0.98,
RMSD = 0.99 cm, and nRMSD (snow depth > 5 cm) = 0.11), different frequency bands (mean r = 0.97, RMSD = 1.46 cm, and nRMSD (snow depth > 5 cm) =0.16), and different GNSS receivers (mean r = 0.62). The data set also has high external consistency with the in-situ measurements and the PMW products, with a consistent

illustration of the interannual snow depth variability. Results from the 17 GNSS sites with the most extended temporal coverage (i.e., from 2013 to 2022) show better performance between GNSS and in-situ that between GNSS and PMW, with RMSD = 2.37 cm & nRMSD = 0.23 for the former, and RMSD = 3.55 cm & nRMSD = 0.35 for the latter. The results also show the good potential of GNSS to derive hourly snow depth observations for better monitoring snow disasters. The proposed framework to develop the data set provides comprehensive and supportive information for users to process raw data of ground GNSS stations with complex environmental conditions and various observation conditions. The resulting GSnow-CHINA v1.0 data set is distinguished from the current point-scale in-situ data or coarse-gridded data, which can be used as an independent data source for validation purposes. The data set is also useful for regional and global climate research and other meteorological and hydrological applications.

Finally, it should be noted that, although we tried our best to reuse the data from the current GNSS networks, there are still limitations concerning the raw data (e.g., limited site numbers and GNSS data types). We look forward to having more sites and data from more GNSS systems (such as from China's Beidou) from the CMA or other organizations to use in the future. Both the algorithm and the data set will be maintained and updated as more years of data become available.

**Author contributions:** WW designed the study and wrote the manuscript. HL provided the GNSS raw data for the production of this data set and co-designed the study. LD provided supportive information for the validation using the PMW snow depth product. LZ provided supportive information for the data filtering. JZ, TY, BL, ZG, and HH contributed to the data/codes accumulation. All authors contributed to the writing and editing of this paper.

**Competing interests:** The authors declare that they have no conflict of interest.

**Acknowledgments:** The first author would like to thank team members from the Meteorological Observation Center, China Meteorological Administration for producing, maintaining, and providing the raw GNSS RINEX data and the in-situ data. The first author also would like to thank team members Ms. Lei Xiao and Ms. Yuan

Gao from Peking University for their contributions to data preparation. The authors would like to thank the

SMAP team, the MODIS team, and the PMW team for archiving and providing the data used in this study. The

first author would like to give her special thanks to Ms. Waner Zhao for her collaboration during the preparation

and writing of this manuscript.

**Financial support:** This study is jointly supported by the National Key Research and Development Program of

China (Grant No. 2019YFE0126600), the National Natural Science Foundation of China (NSFC) projects (Grant

No. 41971377 and No. 41501360), The open fund of the National Earth Observation Data Center (No.

NODAOP2021002), the observing experiment project of Meteorological Observation Center of China

Meteorological Administration (No. SY2020005), and the ESA-MOST China Dragon5 Programme (ID.58070).


**Review statement:** This paper was edited by Dr. Xin Li and reviewed by three referees.

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
