# Peer review of "A new snow depth data set over northern China"

_Earth System Science Data, 2021_

## Author Response (AR1)

**Response to Reviewer #1**

Comment on essd-2021-432

Anonymous Referee #1

The authors proposed a comprehensive framework to process GNSS raw data under complex environment conditions to retrieve snow depth, and based on this, the authors produced a GNSS-based long-term snow depth data set over China from 80 stations. The topic is very interesting from a perspective different from traditional microwave remote sensing retrievals. As the authors' statement, this data set has a unique spatial resolution between point-scale and coarse grid-scale. The new data set is valuable to the science community from this point. I also have confidence in this GNSS-IR technique, which could be a helpful and complementary tool for producing more snow depth products with high spatial-temporal resolution using extended global GNSS networks, particularly from GNSS sites in polar regions or even on scientific expedition vessels. I recommend this work for publication after revisions. Several comments are listed below:

We thank this reviewer for her/his valuable time in reviewing our manuscript and providing thorough and insightful comments. We have carefully revised the manuscript to address the issues and comments raised by the reviewer. Point-to-point responses to the comments are listed below. Comments are shown in **black**, the authors' responses are shown in **blue**, and the revisions in the manuscript are shown in **red**.

Because we have given point-by-point responses during the "open discussion" stage, for the reviewer's convenience, we use "**GREEN color**" to show new responses different from the last version. We have uploaded the revised manuscript using track changes in Word. We have updated all the figures following the three reviewers' comments.

In addition, we have updated the data set during this round to reconsider several issues. The updates are described below. We have also extended the data set to include the recent two snow seasons, i.e., 2020-2021 and 2021-2022. The results show that the quality of the data set has been improved. We have also revised the figures and the corresponding texts in the manuscript to match the updated data set. Some of the updates are shown in the following responses, and the remainings are shown in the revised manuscript.

- Added a new quality flag, i.e., the Signal Strength Indicator (SSI), to do the quality control (SSI >=2).

- Changed the strategy to deal with the non-repeating GLONASS tracks, i.e., used twelve azimuths separated by 30° as a basis to derive the snow-free surface reflector heights.

- Used a more accurate way to consider the penetration depth of the GNSS signal through bare soil, i.e., the penetration depths of each site for GPS L1/L2, GLONASS B1/B2, and BDS B1/B2/B3 were separately calculated using the prepared soil components and VSM parameters.

- Updated the "moving average" method to better filter out the outliers of individual tracks, i.e., using the 12-h window.

- Used the maximum snow depths during 2010-2020 as constraints to remove possible outliers of the raw GNSS snow values per track.

The updated GSnow-CHINA v1.0 data set has been uploaded at https://doi.org/10.11888/Cryos.tpdc.271839.

1. Vegetation and terrain are two significant issues that affect snow depth estimation. The authors only discussed vegetation in Section 5.1. How about terrain effects? I recommend adding in-depth discussions relevant to this issue.

We have added a paragraph to discuss the terrain effects. Please see below:

In practical applications, none of the planar surfaces is entirely horizontal. Small ground tilting angles translate into several tens of centimeters of bias due to the large horizontal distances involved (Larson and Nievinski, 2013). Figure 17 shows simulations for a 2-m antenna height with a variety of snow depth levels and positive terrain slopes using the open-source GPS multipath simulator provided by (Nievinski and Larson, 2014b). For slopes of 5 ° and less, the error in snow depth retrieval is below 10 cm, while for larger slopes (e.g., 8 ° in the figure), the residual effects are ~ 15 cm and higher. Fortunately, for GPS satellites with repeatable ground tracks, such a topographic bias remains stable over time. It thus could be canceled out when using Eq. (1) to estimate snow depth, most of which is the case in this study. While for GNSS satellites like GLONASS and BDS, whose ground tracks are non-repeatable, the terrain effect should be considered. Some previous studies investigated methods to eliminate the influence of terrain (Zhang et al., 2017; Zhang et al., 2020). We are also developing a new approach to consider the terrain effects, which will be demonstrated in a future study.

[Figure]

Figure 18. Simulations of the effects of terrain slopes on snow depth retrievals for a 2-m antenna height of GPS L1 (wavelength = 19 cm)

References:

Larson, K. M. and Nievinski, F. G.: GPS snow sensing: results from the EarthScope Plate Boundary Observatory, GPS Solutions, 17, 41-52, 10.1007/s10291-012-0259-7, 2013.

Zhang, S., Wang, X., and Zhang, Q.: Avoiding errors attributable to topography in GPS-IR snow depth retrievals, Advances in Space Research, 59, 1663-1669, https://doi.org/10.1016/j.asr.2016.12.031, 2017.

Zhang, Z., Guo, F., and Zhang, X.: Triple-frequency multi-GNSS reflectometry snow depth retrieval by using clustering and normalization algorithm to compensate terrain variation, GPS Solutions, 24, 52, 10.1007/s10291-020-0966-4, 2020.

2. Around Line 185: "4) For high- and medium-quality sites, the model for deriving daily reflector height is established, and the raw snow depth for each GNSS satellite, each quadrant, and each GNSS frequency is subsequently calculated as the difference value of the referenced

height in Step 3) and the height of this step". I am confused about the descriptions of "height." Which height was used as the referenced height? The authors should revise the texts to clarify this issue.

We have revised the texts to clarify this issue. Please see below:

1) The observables for snow depth retrieval, i.e., satellite Pseudorandom Noise (PRN) numbers, observation time, satellite elevation angle, satellite azimuth angle, pseudorange, carrier phase (CP), Signal-to-Noise Ratio (SNR), are extracted or calculated from the raw data.

2) The Lomb-Scargle Periodogram (LSP) analysis (Lomb, 1976) is executed on several non-snow days to determine the mean reflector heights for each GNSS satellite, each quadrant, and each GNSS frequency. For those high- and medium-quality sites which will be distinguished in the following Step 3), the mean reflector heights are used as reference heights when calculating snow depth.

3) A comprehensive evaluation of the quality of all the GNSS sites is done based on the data quality of the non-snow surface reflector heights in Step 2), and the sites are divided into high-, medium-, and low-quality accordingly.

4) For high- and medium-quality sites, the model for deriving daily reflector height is established, and the raw snow depth for each GNSS satellite, each quadrant, and each GNSS frequency is subsequently calculated as the difference value of the referenced height in Step 2) and the height of this step.

3. Line 350: "The 8-day MODIS NDVI is also involved as a quality flag in the data set to show the vegetation conditions of the site initially". The authors only gave this vegetation flag. How to use this flag? I recommend adding a few sentences to describe.

We have added descriptions of the NDVI flag in the manuscript. Please see below:

The 8-day MODIS NDVI is also involved as a quality flag in the data set to show the vegetation conditions of the site initially. The 8-day values are combinations of the MODIS MOD13Q1 and MYD13Q1 products. The NDVI flag can provide supplementary information for the users to identify the possible confusion of vegetation. However, due to the coarse resolution of MODIS data, it is not possible to use this flag to represent the actual vegetation cover around the GNSS station.

4. Should Section 4.4 be a separate section? I am afraid it is improper to put the data set descriptions inside Section 4.

We totally agree with the reviewer. We have changed Section 4.4 to a separate section, i.e., "Section 5. Data set descriptions".

5. Figure 17 x-label is not correct? Should it be the current number +10?

The x-label of this figure has been corrected. Also, this figure has been updated as the new Figure 19 in the revised manuscript. Please see below:

[Figure]

**Figure 19.** Examples showing the vegetation effects on snow depth retrieval. The site presented here is bfxc (2015-2020)

6. I am aware that the authors tried their best to reuse the data from the current GNSS networks. There are still limitations concerning the raw data. I look forward to seeing a longer-time series of snow depth products from more sites and systems (such as from China's Beidou). I also encourage the authors to find some way (e.g., making a website?) to maintain and share the methods and data sets to broader users?

We thank this reviewer for helping us considering to improve the potential value of this data set. We totally agree with the reviewer that we should involve more data once they are available in the future. We have added several sentences to address the reviewer's comments at the end of the manuscript. As below:

Finally, it should be noted that, although we tried our best to reuse the data from the current GNSS networks, there are still limitations concerning the raw data (e.g., limited site numbers and GNSS data types). We look forward to having more sites and data from more GNSS systems (such as from China's Beidou) from the CMA or other organizations to use in the future. Both the algorithm and the data set will be maintained and updated as more years of data become available.

In addition, both reviewer #1 and #2 gave us comments on the easy sharing of the data set to a broader international community. We have put the data set on the TPDC website along with this paper which is freely available to the international community (see https://doi.org/10.11888/Cryos.tpdc.271839). We are also considering putting the extended data (e.g., every five years) in the future to some data-sharing websites or making an FTP or website to maintain and share the future data versions.

**Response to Reviewer #2 Prof. Kristine Larson**

Comment on essd-2021-432

Kristine Larson

(1) I think this is an excellent paper describing a new snow depth dataset. My comments below - tagged by page number/line - are meant to improve the readability and value of the paper.

Thanks for all the insightful comments from Prof. Larson to improve the quality of the manuscript. We have revised the contents/figures and given point-to-point responses. Please see below for detailed information. Comments are shown in **black**, the authors' responses are shown in **blue**, and the revisions in the manuscript are shown in **red**.

Because we have given point-by-point responses during the "open discussion" stage, for the reviewer's convenience, we use "**GREEN color**" to show new responses different from the last version. We have uploaded the revised manuscript using track changes in Word. We have updated all the figures following the three reviewers' comments.

In addition, we have updated the data set during this round to reconsider several issues. The updates are described below. We have also extended the data set to include the recent two snow seasons, i.e., 2020-2021 and 2021-2022. The results show that the quality of the data set has been improved. We have also revised the figures and the corresponding texts in the manuscript to match the updated data set. Some of the updates are shown in the following responses, and the remainings are shown in the revised manuscript.

- Added a new quality flag, i.e., the Signal Strength Indicator (SSI), to do the quality control (SSI >=2).

- Changed the strategy to deal with the non-repeating GLONASS tracks, i.e., used twelve azimuths separated by 30° as a basis to derive the snow-free surface reflector heights.

- Used a more accurate way to consider the penetration depth of the GNSS signal through bare soil, i.e., the penetration depths of each site for GPS L1/L2, GLONASS B1/B2, and BDS B1/B2/B3 were separately calculated using the prepared soil components and VSM parameters.

- Updated the "moving average" method to better filter out the outliers of individual tracks, i.e., using the 12-h window.

- Used the maximum snow depths during 2010-2020 as constraints to remove possible outliers of the raw GNSS snow values per track.

The updated GSnow-CHINA v1.0 data set has been uploaded at https://doi.org/10.11888/Cryos.tpdc.271839.

(2) Some of my comments are directed to the figure captions. I think the goal should be to allow people to read/look at the figures without reading the paper. So this means they haev to explicitly say how many sites are in the figures and so on.

We apologize for the incomplete figure captions. We have revised these captions following the corresponding comments. Please see below for detailed information on each revised figure.

(3) My main technical comment is that the authors describe whether they set an azimuth mask

for each site. I do like that they are investigating new ways to evaluate the QC for their sites, but we really need to know a little bit more about how they did that (peak ratios e.g.). I agree that setting the bare soil values is complicate - using NDVI is a good way to get a handle on whether the vegetation is dead and thus a better proxy for bare soil.

Thanks for giving us these valuable technical comments.

We did not use azimuth masks. For each site, we used the "$h_0$ plot" to define rules to evaluate the site quality. Here, $h_0$ is the reference reflector height over non-snow days. The "$h_0$ plot" is made by the sorted $h_0$ values colored by azimuth. One example of the "$h_0$ plot" for site "bttl" is shown as below. If one specific $h_0$ value is within the "flat" segment on the curve, this $h_0$ is treated as a valid value, regardless of which azimuth is. Please also see similar responses to the following comments # 15 & 25.

[Figure]

Figure caption: Example of the "$h_0$ plot" used to define rules to evaluate the quality of each GNSS site. Site name: bttl. Year: 2019

We have added one sentence to describe the "peak ratios" in "Section 3.1 Snow depth retrieval

model" around the lines before Eq. (2): "In this study, the Peak-to-Noise Ratio (PNR) of the LSP is set to be greater than 5 to filter out the quality-controlled satellite tracks."

We have reconsidered the penetration depth of the GNSS signal through bare soil more accurately. We have updated the data set and revised the texts to give detailed descriptions. Please see below:

The penetration depth of the GNSS signal through bare soil ($h_p$) directly influences the determination of the reflector height of the snow-free surface. The $h_p$ is dependent on the soil permittivity and the GNSS wavelength. The soil permittivity is related to soil moisture and soil components. Figure 8 (a) shows the relationship between penetration depth of GPS L1 band and soil moisture/soil components calculated using parameters provided in (Hallikainen et al., 1985). The penetration depth is deeper than 10 cm when soil is very dry (i.e., volumetric soil moisture (VSM) < 0.1 cm$^3$.cm$^{-3}$). The penetration depth is around or shallower than 5 cm under normal soil moisture conditions. In this study, the soil components data for each site, i.e., the percentages of sand and clay, are approximatively derived from the China Soil Science Database (http://vdb3.soil.csdb.cn/) by the soil attributes of the specific city and province that the site is located in. The average VSM of each site is calculated as the multiple-year mean value of the SMAP VSM. The penetration depths of each site for GPS L1/L2, GLONASS B1/B2, and BDS B1/B2/B3 are subsequently calculated using the prepared soil components and VSM parameters. Figure 8 (b) shows the number of GNSS sites categorized by the soil penetration depths ($h_p$). The majority has a shallow penetration depth of 4~8 cm, with only a few having 10 cm or deeper. The $h_0$ is modified as $(h_0 - h_p) + C$ for the final production of the snow depth data set. C is an empirical constant set as 3 cm in this study to represent the offset of the complicated land surface conditions.

[Figure]

**Figure 8.** (a) The penetration depth of GNSS signals over the soil layer, taking GPS L1 band (wavelength = 19 cm) as an example. The red line indicates the mean penetration depth for various soil types; (b) Statistics of the number of GNSS sites categorized by the soil penetration depths (also taking GPS L1 band as an example).

(4) I would also ask the authors to add a few more sentences about how they dealt with the non-repeating ground tracks for Glonass satellites.

In the previous version, since there are only four GLONASS sites in the data set, we used the same strategy as that of GPS to deal with the non-repeating ground tracks. However, it reduced the valid observations. In the revised version, we have changed to use a specific strategy to

reproduce the GLONASS data, using twelve azimuths separated by 30° as a basis to derive the snow-free surface reflector heights.

We have updated the data set. We have also added a paragraph in Section 3.2 to describe how we dealt with the non-repeating ground tracks for GLONASS satellites. Please see below:

For each site, ~ ten days of data with no snow on the ground are used to calculate the raw snow-free surface reflector height ($h_0$). According to the data availability, days of the year (DOYs) 110~119 or DOYs 274~283 are generally selected since these days have no snow according to historical in-situ data. Specifically, for GLONASS, to deal with the non-repeating tracks, one-month snow-free data (DOY 105~135) are used to calculate the raw $h_0$. The reflector height for each GNSS satellite, quadrant, and GNSS frequency band is calculated using the Lomb-Scargle spectrum, and it is just the initial height being used for the quality evaluation of the GNSS sites. Due to the complex natural environment for various sites, it is not clear whether one site is suitable for snow depth retrieval. The following section will define a rigorous rule to evaluate the quality of all the GNSS sites. For those high- and medium-quality sites determined in the following section which are suitable for snow depth retrieval, the finalized snow-free surface reflector height will be determined as the mean value of heights of the ten days.

It is worth mentioning that GPS ground tracks have sidereal repeatability and reappear at the same azimuth. In contrast, GLONASS satellite and BDS MEO satellite have non-repeating ground tracks. GLONASS orbits repeat every eight sidereal days, with the ground track shifted by 45° in longitude per day (Tabibi et al., 2017b). BDS MEO satellites repeat approximately every seven sidereal days (Ye et al., 2015). In this study, there are only 4 GLONASS sites (i.e., bfqe, bttl, hltl, and hlhl) and 1 BDS site (e.g., qxdw). The strategy for processing GLONASS data is slightly different from that of GPS, i.e., the snow-free surface reflector heights are given in twelve azimuths separated by 30° for all available GLONASS satellite tracks and frequency bands. While for BDS satellite, due to the relatively low number of available satellites, the reflector height is given by quadrant only without distinguishing tracks and frequency bands to preserve as many observations as possible. Previous research developed a multistep clustering algorithm to handle the non-repeating ground tracks of GLONASS (Tabibi et al., 2017a). We are also developing a new algorithm in an upcoming study considering terrain effects, which will be particularly effective for non-repeating tracks.

References:

Tabibi, S., Nievinski, F., and Van Dam, T.: Statistical Comparison and Combination of GPS, GLONASS, and Multi-GNSS Multipath Reflectometry Applied to Snow Depth Retrieval, 10.1109/TGRS.2017.2679899, 2017b.

Ye, S., Chen, D., Liu, Y., Jiang, P., Tang, W., and Xia, P.: Carrier phase multipath mitigation for BeiDou navigation satellite system, GPS Solutions, 19, 545-557, 10.1007/s10291-014-0409-1, 2015.

(5)  I do echo the comment of the previous reviewer. For this dataset to be truly useful, it needs to be easily found. For soil moisture, I could point to the International Soil Moisture Netwrok. Even though my soil mositure project has ended, the ISMN has provided a way for researchers to use our soil moisture data without contacting me directly. I do not know the best place for snow data, particularly for the international community.

We have put the data set on the TPDC website along with this paper which is freely available to the international community (see https://doi.org/10.11888/Cryos.tpdc.271839). We are also considering putting the extended data (e.g., every five years) in the future to some data-sharing websites or making an FTP or website to maintain and share the future data versions.

First page

(6) A new snow depth data set over northern China by developing a comprehensive framework using the complex GNSS station network

I would suggest a slightly different title. We don't use complex in quite this way. Yes, the dataset is complicated - but really the issue is that the sites were not located in an ideal way for snow sensing. Maybe:

"A new snow depth data set over northern China derived using GNSS interferometric reflectometry from a continuously operating GNSS network."

This is a suggestion - not any kind of required change.

Thank you for your suggestion. We took your advice and use this title "A new snow depth data set over northern China derived using GNSS interferometric reflectometry from a continuously operating network".

(7) page 3, line 67 I think you should explicitly say in the US (the reader should not need to go to the website to find out that SCAN and SNOTEL are only in the western US).

We have added "In-situ measurements from ground networks such as SCAN and SNOTEL in the United States"

(8) page 4, line 92 For **typical** GNSS sites the spatial footprint is ~1000 m^2. (rather than saying recognized).

We have corrected "For typical GNSS sites the spatial footprint is ~1000 $m^2$, which is a scale between point-scale and satellite-scale"

(9) page 4 last couple sentences my copy has the commas in 9,000,000 and 4,200,000 as ' rather than ,

We have corrected "China's annual mean snow extent is greater than 9,000,000 $km^2$, with a stable snow-covered area of ~ 4,200,000 $km^2$."

page 6, line 138

(10) and the stations have turned into a certain amount since 2012.

better to say " and the station build phase was completed in 2012. " ???

We have rephrased. "China started to construct ground GNSS stations in 2009, and the station build phase was initially completed in 2012 with some regions later in 2015."

page 7

(11) I think it is sufficient to say you used the broadest ephemeris. They should be the same everywhere and it should not matter where you got them from.

We have simplified this sentence "The broadcast ephemeris was used to calculate each GNSS

satellite's position."

(12) page 7, last line. the RINEX files may truly only have 10 degree data - but was that imposed when the file was made or when the GPS station was set up? It is pretty unusual to set that at the RINEX creation stage (not saying it wasn't done, just asking to be clear).

We have checked with the CMA data administrator and rewritten this sentence. For CMA and CEA sites, the minimum elevation angle of the GNSS satellite was set to be 10° when the sites were built.

(13) Shouldn't there be references for these data, SMAP, NDVI data etc?

The references have been added. As below:

O'Neill, P. E., Chan, S., Njoku, E. G., Jackson, T., Bindlish, R., and Chaubell, J.: SMAP L3 Radiometer Global Daily 36 km EASE-Grid Soil Moisture (Version 6), NASA National Snow and Ice Data Center Distributed Active Archive Center [dataset], https://doi.org/10.5067/EVYDQ32FNWTH, 2019.

Didan, K.: MODIS/Terra Vegetation Indices 16-Day L3 Global 1km SIN Grid (V061), NASA EOSDIS Land Processes Distributed Active Archive Center (LP DAAC) [dataset], https://doi.org/10.5067/MODIS/MOD13A2.061, 2021.

(14) Add a reference for Lomb Scargle Periodogram

The reference has been added. As below:

Lomb, N. R.: Least-squares frequency analysis of unequally spaced data, Astrophysics and Space Science, 39, 447-462, 10.1007/BF00648343, 1976.

(15) As part of your quality control, I think you have set a mask for each site, but you don't really say it. Why not make it clear? You should not bother calculating reflector heights if they are not useful for snow sensing.

We did not use azimuth masks. The main form of the defined rule is the "$h_0$ plot". An example of the "$h_0$ plot" was shown in Comment # (3) mentioned above, and also is shown at the bottom of each subfigure in Figure 6 (copied below). The "$h_0$ plot" shows sorted $h_0$ values colored by azimuth. If one specific $h_0$ value is within the "flat" segment on the curve, this $h_0$ is treated as a valid value, regardless of which azimuth is.

In addition, we have written the last two paragraphs of "Section 3.3 Quality evaluation of the GNSS sites" to make the definition of the rule clearer. Please see detailed revisions in the following Comment # (22).

[Figure]

**Figure 6.** Examples show the high/medium-quality sites and the low-quality sites. High/medium-quality sites: (a) bumz, 2017; (b) bfhr, 2019; (c) bgfc, 2019; Low-quality sites: (d) uqwl, 2019; (e) qhdl, 2020; (f) qhbm, 2018. The top image in each subfigure shows the footprint of the observation for elevation angles of 10°, 15°, 20°, 25°, and 30°, respectively. The bottom image in each subfigure shows the distribution of the reflector heights for non-snow surfaces calculated from 10-days of observations using the SNR model. The background of this figure is from Google Earth (https://earth.google.com/web/) © Google Earth 2021.

page 9

(16) mean Peak-to-Noise Rate (PNR). I think you mean peak to noise ratio, not rate

We have corrected "In this study, the Peak-to-Noise Ratio (PNR) of the LSP is set be greater than 5 to filter out the quality-controlled satellite tracks."

page 11

(17) line 219 are analyzed in Table 1, You really mean to say that they are listed - not analyzed

We have corrected "The main formulas and applicability of the five models mentioned above to the data of GNSS sites in this study are listed in Table 1"

(18) Even though, the SNR model has been verified to have higher accuracy than the L4 and F2C models (Liu et al., 2021).

remove "Even though"

We have removed "Even though". "The SNR, L4, and F2C models are suitable for all sites because the observables used as inputs for these models are available in the GNSS raw data.

, The SNR model has been verified to have higher accuracy than the L4 and F2C models (Liu et al., 2021)."

(19) It is worth mentioning that, for GPS and GLONASS satellite, the reflector height is given per satellite, quadrant, and frequency band, while for BDS satellite, the reflector height is given by quadrant only because the BDS MEO satellite changes its trajectory day by day.

GLONASS does not have a daily repeatable ground track. What do you do to account for this?

We have reproduced the GLONASS data using a new strategy described in Comment # (4). We have revised these sentences and added a paragraph in the same place in this Section 3.2 to describe the method that we dealt with the non-repeating ground tracks for GLONASS satellites. Detailed revisions have been shown in Comment # (4). To save the reviewer's time, we will not repeat here.

Figure 13

(20) The caption of Figure 4 should be expanded to describe the subplots, especially the ones with data in them. I know what is in the figures, but most people would not.

I also suggest using grid lines in the plots.

We have expanded the caption of Figure 4, and we have also added grid lines in the plots. Please see below:

[Figure]

**Figure 4.** Geometry and principle of the SNR model. (a) The geometry of the direct and reflected signal over the snow surface; (b1) Example of the recorded GNSS SNR data and the removal of the direct signal with a second-order polynomial; (b2) Residual of (b1) below elevation angle (E) of 30°, converted from dB to linear units (for simplicity, Volts); (b3) Lomb-Scargle analysis of (b2) to find out the dominant frequency of the transformation and the resulting reflector height.

(21) The majority of the CEA antennas are settled down on a standard rooftop, with the GNSS receivers being put in the accompanying small house. It explains why most of the CEA sites are not suitable for snow depth retrieval.

I suggest you say "set upon a rooftop." Settled is used for something that starts at one height and slowly changes, like snow settling.

We have revised this sentence. "The majority of the CEA antennas are set upon a rooftop (e.g. Site qhdl in Figure 5), with the GNSS receivers being put in the accompanying small house. It explains why most of the CEA sites are not suitable for snow depth retrieval."

(22) Figure 6 shows the rule being applied to six individual sites with various surroundings,

You say this before you define the rule. I think you really need to define the rule first.

We have written the last two paragraphs of "Section 3.3 Quality evaluation of the GNSS sites" to clarify this issue. Plese see below:

A rigorous rule is defined to evaluate the quality of all the GNSS sites. For each site, the 10-day reflector heights of non-snow surface (i.e., $h_0$) are calculated, sorted, and colored by azimuths to make a "$h_0$ plot". Examples of the "$h_0$ plot" are shown at the bottom of each subfigure in Figure 6. The "$h_0$ plot" is visually checked carefully and determines whether it is suitable for the retrieval of snow depth. Suppose one site shows relatively long and stable $h_0$ values during the entire observation period. In that case, i.e., the "$h_0$ plot" has a relatively "flat" segment on the curve, which indicates that this site is qualified to determine the initial range of the non-snow surface reflector height. Afterward, a range of $h_0$ is given manually to narrow the good $h_0$ values. The difference of the minimum and maximum value of the range is set to be no more than 0.5 m. The finalized non-snow surface reflector height for each satellite, each quadrant, and each GNSS frequency are respectively determined as the mean value of the good heights of the ten days. In contrast, if one site has no "flat" segment on the "$h_0$ plot", this site is determined as a low-quality site and will not be used for snow depth retrieval. It should be noted that during this processing step, it can only eliminate those sites with poor data quality for snow depth retrieval rather than distinguishing high- and medium-sites. There are no apparent differences for the high- and medium-quality sites regarding the natural environment. Instead, the medium-quality site is defined using two simple rules, i.e., one is the site has good-quality data, but there is no snow for almost all the years. The other is the site's lack of data for most of the years.

Figure 6 shows the defined rule applied to six individual sites with various surroundings, i.e., bumz, bfhr, bgfc, uqwl, qhdl, and qhbm. The top panel of each subfigure shows the environmental conditions around the station on Google Map, with different colors indicating the footprints for elevation angles of 10°, 15°, 20°, 25°, and 30°, respectively. The bottom panel of each subfigure shows the sorted 10-day reflector heights of non-snow surface (i.e., $h_0$). The plots clearly show the differences in the heights for different sites. The first two sites, i.e., bumz and bfhr, show relatively long and stable $h_0$ values for all the GNSS satellites, quadrants, and frequency bands during the entire observation period. It indicates that these sites are flat enough for all the orientations and are ideal for determining the initial range of the non-snow surface reflector height, i.e., 2.5 ~ 2.8 m for bumz and 2.8 ~ 3.1 m for bfhr. Unlike these two sites, the

bgfc site has relatively stable $h_0$ values only in specific orientation whose natural condition is open and flat; This phenomenon can be verified from the photo of the site in Figure 5. This site is also good enough to determine the initial range of the non-snow surface reflector height, i.e., $3.6 \sim 4.1$ m for bgfc. On the contrary, the three sites at the bottom of Figure 6, i.e., uqwl, qhdl, and qhbm, show continuously changed $h_0$ values because of the poorly defined peaks for most Lomb-Scargle periodograms. It indicates that it is unreliable to determine a true $h_0$ due to complex environmental conditions.

Figure 6

(23) How do we know that these are good retrievals? Maybe some of the periodograms have poorly defined peaks?

We agree with this reviewer. We have added one sentence to explain this issue. As below:

On the contrary, the three sites at the bottom of Figure 6, i.e., uqwl, qhdl, and qhbm, show continuously changed $h_0$ values because of the poorly defined peaks for most Lomb-Scargle periodograms. It indicates that it is unreliable to determine a true $h_0$ due to complex environmental conditions.

page 16

(24) the three sites at the bottom of Figure 6, i.e., uqwl, qhdl, and qhbm, show continuously changed hX values. It indicates that it is unreliable to determine a true hX by the Lomb-Scargle spectrum due to complex environmental 280 conditions.

I know what you are trying to say here, but I think it is simply a matter that you are computing reflector heights at sites where you should not bother. The sites are surrounded by too much clutter.

We agree with this reviewer that these sites are not suitable for snow depth retrieval. However, initially, we did not know the environmental condition of each site because the CMA who provided the raw GPS data did not give us photos or other supportive information. The only information we had was the Google map; This is why we defined the rule mentioned above (see Comment # (22)) to pick out these "bad" sites.

(25) When you are using your rules to find good sites, do you prepare and save azimuth masks? It is not clear to me that you do.

To save your time, the response to this comment has been shown in Comment # (15).

page 18

(26) 0 ~ 12 am or 12 ~ 24 pm within one specific day.

You should say 0-12 UTC and 12-24 UTC.

We have corrected. "For most sites with only GPS observations, we try to produce 12-hour snow depth if there are no less than five valid observations from $0 \sim 12$ UTC or $12 \sim 24$ UTC within one specific day."

(27) Figure 7 is this for one site? Which one?

Yes, this is for one site named "bfqe". We have added the site name in the figure caption. We have updated the values in this figure using the revised data set, and we have also revised the

style of this figure to better show the outliers. As below:

[Figure]

**Figure 7.** Examples showing the moving-average filtering of the snow depth results over one snow season. The site presented in this figure is "bfqe" which is a CMA site. DOY: day of year.

(28) Figure 9 add the number of sites this represents in the caption. My recollection is that you don't have that many that observe both GPS and Glonass.

We have added the number of sites in the caption. There are only 4 sites that observe both GPS and GLONASS. We have updated the values in this figure using the revised data set. According to reviewer #3's comments, we have also revised the style of this figure to show the correlations of data with snow depths greater than 5 cm. As below:

[Figure]

**Figure 9.** Correlations of 24 h/12 h snow depths from GPS and GLONASS observations. (a) 24 h; (b) 12 h. The error bar of each point is the standard error (STE) of the snow depths for all the observation records. Four available sites, i.e., "hltl", "hlhl", "bfqe", and "bttl", during the GPS/GLONASS overlapped periods (i.e., the year 2014 and 2015) are used to plot this figure. For each point in the figure, the number of valid observations is more than five. To prevent other possible effects besides the GNSS system, the STE of snow depths is less than 1 cm. Blue points are with the retrieved GPS and GLONASS snow depths greater than 5 cm. RMSD: Root Mean Square Difference; rRMSD: relative RMSD.

(29) Figure 10- Couldn't part of the difference between L1 and L2 be due to the phase centers not being in the same place? Did you assume the phase centers were the same? How many sites are shown in these figures?

We did not consider the L1/L2 difference of the antenna phase centers. We agree with the

reviewer that the raw reflector heights of L1 and L2 have bias. For snow depth in this study, we calculated the L1 and L2 snow depth separately, i.e., using h0_L1-h_L1 and h0_L2-h_L2. Here h0 and h are the reference snow-free height and the daily height. Because we subtracted the bare soil value, the L1/L2 bias no longer exists (or is very small). We have added discussions on this issue in the revised manuscript. Please see below:

Figure 10 (a1, a2) and (b1, b2) shows correlations of the snow depths between GPS L1 and L2 and between GLONASS L1 and L2, respectively, using data from the same four GPS/GLONASS compatible sites as in Figure 9. The results from different frequency bands show good consistency with each other, with r = 0.91, RMSD = 2.90 cm, and rRMSD = 7.27% for GPS, and r = 0.98, RMSD = 1.84 cm, and rRMSD = 5.93% for GLONASS (Figure 10 (a1) and (b1)). The RMSD (rRMSD) values of snow depths greater than 5 cm are 2.68 cm (8.83%) for GPS and 1.86 cm (8.42%) for GLONASS. It should be noted that a small part of the difference between L1 and L2 is due to the antenna phase centers not being in the same place. The initial bias occurs on the raw L1 and L2 reflector heights. However, the final bias becomes negligible because, during snow depth calculation, the reflector height value of bare soil is subtracted. The BDS results still are not used for comparison due to the limited number of observations.

We have added site numbers used to make these figures. We have updated the values in this figure using the revised data set. According to reviewer #3's comments, we have also revised the style of this figure to show the correlations of data with snow depths greater than 5 cm.

The finalized figure is shown below:

[Figure]

**Figure 10.** Correlations of snow depth from different GNSS frequencies. (a1) GPS L1 vs. GPS L2; (b1) GLONASS L1 vs. GLONASS L2. The color bar represents the density of points; (a2) Same as (a1) but with snow depths greater than 5 cm; (b2) Same as (b1) but with snow depths greater than

5 cm. Fifty-one high-quality GPS sites of CMA and four GPS/GLONASS compatible sites are respectively used to plot (a1, a2) and (b1, b2) of this figure. The error bar of each point in (a2) and (b2) is the standard error (STE) of the snow depths for all the observation records. For each point in the figure, the number of valid observations is more than five. To prevent other possible effects besides the GNSS frequency, the STE of each snow depth is less than 3 cm in (a1) and (b1) and less than 1 cm in (a2) and (b2). RMSD: Root Mean Square Difference; rRMSD: relative RMSD.

(30) Figure 11 add the 1 to 1 (diagonal) line as you had in Figure 10.

How many sites are represented in each figure? This information should be in the caption or in the figure.

We have added the 1 to 1 line. We have revised the caption to include the number of sites. We have updated the values in this figure using the revised data set. According to reviewer #3's comments, we have also revised this figure to show the relative Standard Errors (rSTEs) for each type.

The finalized figure and the revised texts are shown below:

[Figure]

**Figure 11.** Comparisons of the GNSS-derived snow depth and the in-situ measurements from different types of GNSS receivers: (a1) Trimble; (b1) Leica; (c1) Minshida (MSD), and the histogram of the standard error (STE) and relative STE (rSTE) of snow depths for different types of GNSS receivers: (a2) Trimble; (b2) Leica; (c2) Minshida (MSD). The number of sites representing Trimble, Leica, and MSD is 20, 5, and 24. To prevent other possible effects besides the receiver type, each data point used to plot this figure is more than ten valid observations, and the STE of snow depths is less than 1 cm in (a1), (b1), and (c1). In (a2), (b2), and (c2), the GNSS snow depths values are greater than 5 cm.

The CMA and CEA sites are set up with various brands of GNSS receivers. Most of these receivers are from three brands, i.e., Trimble, Leica, and MinShiDa (MSD). Taking these three brands as examples, in order to evaluate the snow depth results from these three brands, Figure 11 (a1), (b1), & (c1) respectively show the differences of the snow depths derived from the three brands, taking the in-situ measurements as benchmarks. The results from the three brands show good consistency with r = 0.87, 0.92, and 0.88, respectively. Figure 11 (a2), (b2), & (c3) further show the histogram of the STEs and relative STEs (rSTEs) of the snow depths from the three brands, and good consistency is also shown in these subfigures. The maximum of the statistical STE (rSTE) for Trimble, Leica, and MSD is respectively around 1 cm (7%), 0.6 cm (4%), and 1 cm (7%).

page 23

(31) In addition, the peak of the PMW snow trend for each snow season moves to the right,

Say instead that the peak is later in the season rather than moves to the right

We have corrected. "In addition, the peak of the PMW snow trend for each snow season is later in the season, which is due to the change of snow grain size (Dai et al., 2012)."

(32) Figure 12 Explicitly say in the caption how many sites are shown.

We have added the number of sites in the caption. We have updated the values in this figure using the revised data set. According to reviewer #3's comments, we have also revised this figure to show the RMSDs and relative RMSDs.

The finalized figure is shown below:

[Figure]

**Figure 13.** Comparisons of daily mean snow depth derived from GNSS, in-situ, and PMW for17 GNSS sites with the most extended temporal coverage (i.e., from 2013 to 2022). RMSD: Root Mean Square Difference; rRMSD: relative RMSD.

page 24

(33) This result indicates that the laser measurements in operational meteorological observations are not always reliable.

Could say "this result is a reminder that operational laser measurements of snow depth are not always reliable.

We have corrected. "This data point is an outlier because the historical weather reports showed no significant snowfall events before or after these dates. This result is a reminder that operational laser measurements of snow depth are not always reliable."

page 30

(34) First, the minimum elevation angle of GNSS satellites should be set to 5 °~ 15 °to preserve

the multipath effect as much as possible because only data with low elevation angles can show the surface reflection.

I don't understand this sentences. If you had said "set to 5 degrees" I would agree because you said you are setting a minimum, which should be a single number. I do not know why you mention 15. Despite the fact that these organizations unfortunately used an elevation mask of 10 degrees, your paper demonstrates that stations with an elevation minimum of 10 degrees can be used (though not optimal).

We apologize for the unclear expression. We tried to say that, under normal circumstances, the minimum elevation angle of 5°, 10°, and 15° should be all fine. However, we did not test other combinations, like 15°-35°, 15°-30°, etc. We have revised this sentence as below:

"First, the minimum elevation angle of GNSS satellites should be set to a single number like 5° or 10° to preserve the multipath effect as much as possible because only data with low elevation angles can show the surface reflection."

(35) Third, the cycle slip of GNSS observation can severely reduce the data quantity available for snow depth retrieval.

I don't understand why a cycle slip would by itself reduce the data quantity for SNR?

We apologize for misunderstanding the meaning of cycle slip. We have replaced this sentence using the following to clarify this issue:

Third, the GNSS tracks may miss data in some epochs during the ascending or descending sequences, although they satisfy the condition of minimum to maximum elevation angles. These data are removed in this study to ensure the accurate acquisition of the reflector heights.

page 31

(36) Also, the snow depth results on snowy days could, to some extent, affect the accuracy.

What do you mean here? I don't disagree with you - but if you say it, you have to explain why you say it.

We are sorry for the unclear description. We wrote this because we read this point from the conclusions of the following paper. However, we misunderstood the real intention of the authors. For simplicity, we have deleted this sentence.

Anonymous Referee #3

**General comments:**

The authors present a new snow depth dataset using a network of GNSS stations in northern China. The data were derived using a varied of established methods depending on data quality and instruments available at the different stations. They furthermore propose a method to automatically control the data quality.

The quality of the obtained snow depth data is evaluated comparing the different GNSS snow derivation methods with each other and to in situ manual measurements and passive microwave data.

The presented dataset is a valuable contribution to the research community and the paper should be accepted for publications after the following points are taken into consideration.

Thanks for all the constructive comments from Referee #3 to improve the quality of the manuscript. We have revised the contents/figures and given point-to-point responses. Please see below for detailed information. **Because we have given point-by-point responses during the "open discussion" stage, the version below is the same as that we uploaded previously.**

During this round of revision, we have uploaded the revised manuscript using track changes in Word. We have updated all the figures following the three reviewers' comments.

Comments are shown in **black**, the authors' responses are shown in **blue**, and the revisions in the manuscript are shown in **red**.

In addition, we have updated the data set during this round to reconsider several issues. We have also extended the data set to include the recent two snow seasons, i.e., 2020-2021 and 2021-2022. The results show that the quality of the data set has been improved. We have also revised the figures and the corresponding texts in the manuscript to match the updated data set. Some of the updates will be shown in the following responses, and the remaining will be shown in the revised manuscript during the subsequent "final response" stage.

The updates are summarized as follows:

- Added a new quality flag, i.e., the Signal Strength Indicator (SSI), to do the quality control (SSI >=2).

- Changed the strategy to deal with the non-repeating GLONASS tracks, i.e., used twelve azimuths separated by 30° as a basis to derive the snow-free surface reflector heights.

- Used a more accurate way to consider the penetration depth of the GNSS signal through bare soil, i.e., the penetration depths of each site for GPS L1/L2, GLONASS B1/B2, and BDS B1/B2/B3 were separately calculated using the prepared soil components and VSM parameters.

- Updated the "moving average" method to better filter out the outliers of individual tracks, i.e., using the 12-h window.

- Used the maximum snow depths during 2010-2020 as constraints to remove possible outliers of the raw GNSS snow values per track.

The updated GSnow-CHINA v1.0 data set has been uploaded at https://doi.org/10.11888/Cryos.tpdc.271839.

(1) The RMSD (Figure 9 and 10) and STE (Figure 11) should be given also as relative errors (%), allowing a better comparison to other studies. This is particular important since the snow depth in the studied area is particularly low compared to other snow-covered areas in the world. Moreover, data points with < 5cm were included in analysis and seems to be the majority of the datapoints (See figure 10). However, later it was stated that the obtained results are not reliable for snow depth < 5 cm (line 493). I see a contradiction here. Due to the high density of points with very low snow depth the reported deviations can be misleading. Consider also providing the analysis only for datapoints above a certain threshold. It would be also useful to add a regression line to the scatter plots in figures 9 to 11.

We have updated Figures 9, 10, and 11 to include relative errors (%). We have provided the analysis only for data points above a certain threshold (i.e., 5 cm). We have added regression lines to the scatter plots in Figures 9 and 11.

We have revised the texts to match the revised figures. We have also added one new sub-section, i.e., "Section 3.6 Error indicators used in this study" in the "Methods" section, to define the four error indicators used: RMSD, rRMSD, STE, and rSTE.

Please see below for detailed information:

**4.1 Intra-comparisons of GNSS snow depth results**

[revised manuscript text omitted]

(2)    It would be nice if more recent data would be included in the dataset.

We have extended the data set to include data from the recent two snow seasons, i.e., 2020-2021 and 2021-2022. We have also updated all the figures in the manuscript to match the updated data set.

(3)    The site info are given only in .doc an .xls formats. It would be good to have that

information also in non-proprietary format as text file or csv.

We have provided the TXT and CSV versions in the revised data set.

**Detailed comments:**

(4)   Line 153-155: I don't understand what the authors mean with: " … we preserved the high-quality and medium-quality sites as much as possible…"

We have rewritten this sentence to make it clearer. "Each GNSS site has irreplaceable value because of its unique natural environment and characteristic of snow. Therefore, regardless of the raw data incompleteness in some periods for some sites, we preserve the high-quality and medium-quality sites as much as possible during the production of the data set."

(5)   Figure 5: Please indicate which sites are considered high quality or low quality in figure caption.

We have added descriptions in the figure caption. "Figure 5. Photos of typical GNSS sites. "bumz" and "bgfc" are two high-quality CMA sites, and "qhdl" is a low-quality CEA site that is not suitable for snow depth retrieval."

(6)   Line 276: Please explain why the site bgfc is has stable h0 only at specific orientations.

We have added explanations relative to this issue. "Unlike these two sites, the bgfc site has relatively stable $h_0$ values only in specific orientation whose natural condition is open and flat. At the same time, it is impossible to derive correct $h_0$ values in other orientations that have buildings or trees; This phenomenon can be verified from the photo of the site in Figure 5."

(7)   Line 284: a value range of 0.5 m is still pretty large if it is considered that the snow depth to be measure is generally lower than 0.3 m. Can you comment on this point?

The value range of 0.5 m is just a rule to define valid $h_0$ values and eliminate the azimuths that are not suitable for snow depth retrieval. The GPS tracks are repeatable and appear at the same azimuth each day, and the snow depth is calculated using $h_{snow} = h_0 - h$. Therefore, the differences in heights from other azimuths do not affect the snow depth results. Also, for non-repeatable GLONASS tracks, we used twelve azimuths separated by 30° as a basis to derive the snow-free surface reflector heights, which effectively eliminated the terrain effects.

(8)   Line 316: Specify the length of the moving window. 10 datapoints, hours, days, …?

We have added descriptions. "Snow depth values out of the 95% confidence interval are smoothed over a sliding window across neighboring elements. The length of the moving window is set to be 12-hour in this study."

We have also remade Figure 7 to match the updated "moving average" method. The new Figure 7 is as below:

[Figure]

**Figure 7.** Examples showing the moving-average filtering of the snow depth results over one snow season. The site presented in this figure is "bfqe" which is a CMA site. DOY: day of year.

(9) Figure 7: The oscillations/noise are much bigger than the reported errors. Can you comment on this?

Thank you for pointing out this issue. For snow depth retrieval using GNSS data, the snow depth accuracy is highly related to the correct recognition of the peak frequency on the Lomb-Scargle spectrum. The peak frequency is directly used to convert to surface height. However, although we have defined rigorous rules to derive high-quality Lomb-Scargle spectrums, there is still wrong recognition of peaks, perhaps due to the unknown environmental abnormity. One example of this issue is shown in the following figure. Therefore, it is necessary to remove these outliers using moving average methods. Also, the accuracy for daily or hourly results is ensured by computing the mean value of adequate observations (e.g., > 5 in this study).

[Figure]

**Figure.** Examples of correct (left) and incorrect (right) peaks on the Lomb-Scargle Spectrum. Left: DOY 300 for site "bfqe", PRN 15, Quadrant 4, and GPS L1; Right: DOY 300 for site "bfqe", PRN 29, Quadrant 1, and GPS L1. For the right figure, the correct peak should be the one next to the recognized peak. The incorrect peak in this example leads to a height estimate error of (3.24-3.18)*100=6 cm.

We have added explanations to this issue in the revised manuscript. "For each site, as shown in Figure 7, the raw snow depth values over a snow season, i.e., from October 1st this year to April 30th the following year, are gathered together. The moving average algorithm is executed to filter out the snow depth outliers probably due to the incorrect recognition of the peak frequencies on the Lomb-Scargle spectrums."

(10) Lines 321-340: Is the volumetric soil moisture not changing over the year? How is this accounted in the data derivation. What is the effect of soil freezing?

The soil moisture only contributes to the reflector height estimation of the snow-free surface.

When there is snow on the ground, the height (or saying the peak frequency of the Lomb-Scargle spectrum) represents "specular" reflections from the surface of the snow. In other words, for the snow depth estimates using GNSS-IR technique, it is assumed the interactions of multiple snow layers and the underlying soil do not affect the result. Therefore, soil freezing does not affect the result as well. Also, we did not consider the change in the volumetric soil moisture (VSM) over the year. Instead, we use the multiple-year mean VSM as inputs in the model to account for the effect when computing the referenced height ($h_0$) of snow-free surface. A multiple-year mean can give a relatively stable reflection of the status of soil for a specific site.

In the revised data set, we have updated it to use a more accurate way to consider soil penetration depth. We use the multiple-year mean SMAP VSM and the soil components as inputs to calculate the penetration depth of the individual site. We have also revised the figure and texts in Section 3.5 (3). Please see below for detailed information.

(2) Modifying the system errors caused by the penetration depth of soil

The penetration depth of the GNSS signal through bare soil ($h_p$) directly influences the determination of the reflector height of the snow-free surface. The $h_p$ is dependent on the soil permittivity and the GNSS wavelength. The soil permittivity is related to soil moisture and soil components. Figure 8 (a) shows the relationship between penetration depth of GPS L1 band and soil moisture/soil components calculated using parameters provided in (Hallikainen et al., 1985). The penetration depth is deeper than 10 cm when soil is very dry (i.e., volumetric soil moisture (VSM) < 0.1 cm$^3$.cm$^{-3}$). The penetration depth is around or shallower than 5 cm under normal soil moisture conditions. In this study, the soil components data for each site, i.e., the percentages of sand and clay, are approximatively derived from the China Soil Science Database (http://vdb3.soil.csdb.cn/) by the soil attributes of the specific city and province that the site is located in. The average VSM of each site is calculated as the multiple-year mean value of the SMAP VSM. The penetration depths of each site for GPS L1/L2, GLONASS B1/B2, and BDS B1/B2/B3 are subsequently calculated using the prepared soil components and VSM parameters. Figure 8 (b) shows the number of GNSS sites categorized by the soil penetration depths ($h_p$). The majority has a shallow penetration depth of 4~8 cm, with only a few having 10 cm or deeper. The $h_0$ is modified as $(h_0 - h_p) + C$ for the final production of the snow depth data set. C is an empirical constant set as 3 cm in this study to represent the offset of the complicated land surface conditions.

[Figure]

**Figure 8.** (a) The penetration depth of GNSS signals over the soil layer, taking GPS L1 band (wavelength = 19 cm) as an example. The red line indicates the mean penetration depth for various

soil types; (b) Statistics of the number of GNSS sites categorized by the soil penetration depths (also taking GPS L1 band as an example).

(11)Line 354: Please indicate relative to which quantity the standard error is calculated?

We have added descriptions. "For each snow depth data record, the STE of the snow depths for different satellite tracks is treated as another qualifying flag."

(12)Figure 12: I find quite unusual to compare the seasonal evolution of the snow depth only for the mean of the 17 sites since in this way oscillations and outliers are probably smoothed away. It would be good to see also the comparison of the 3 methods for single sites, which would give an indication of the validity of the data in the single cases.

We have added comparisons of the three methods for single sites in the new Figure 12. We have also rewritten all the texts related to the current Figure 12 and Figure 13 in the revised manuscript. Please see below:

Figure 12 shows an example of the comparisons of daily snow depth derived from GNSS, in-situ, and PMW. The data used in this figure is from 16 GNSS sites in 2016-2022, with the least missing daily snow depth values. The comparison period is from 2016 to 2022 due to the data discontinuity in other periods. The three data sets have similar variation trends but with apparent differences in absolute snow depth values. The GNSS-derived snow depths are closer to the in-situ values than the PMW for most sites because GNSS and in-situ have a closer footprint. However, for some sites (e.g., Site "jldg" in Figure 12), the in-situ measurements are much higher than the GNSS and PMW, which needs further in-depth analysis. Figure 12 presents all the GNSS snow depth values derived in the produced GSnow-CHINA data set, regardless of its quality, to give a comprehensive illustration of the data. It is recommended that the users define their own rules to determine whether to use those snow depth values with low numbers of GNSS tracks or high STE.

[Figure]

**Figure 12.** Comparisons of daily snow depth derived from GNSS, in-situ, and PMW. The data used in this figure is from 16 GNSS sites in 2016-2022, with the least missing daily snow depth values.

Figure 13 shows an example of the comparisons of daily mean snow depth derived from GNSS, in-situ, and PMW. The data used in this figure is from 17 GNSS sites with the most extended temporal coverage (i.e., from 2013 to 2022). As expected, the GNSS and in-situ data have similar performance compared to the PMW data, with RMSD = 2.08 cm & rRMSD = 10.40% for GNSS vs. in-situ, and RMSD = 3.53 cm & rRMSD = 17.68% for GNSS vs. PMW. In addition, the peak of the PMW snow trend for each snow season is later in the season, which is due to the change of snow grain size (Dai et al., 2012).

[Figure]

**Figure 13.** Comparisons of daily mean snow depth derived from GNSS, in-situ, and PMW for17 GNSS sites with the most extended temporal coverage (i.e., from 2013 to 2022). RMSD: Root Mean Square Difference; rRMSD: relative RMSD.

(13) Figure 13: It would be good to indicate the standard deviation or RMSE for the mean and max values.

We have added the RMSDs in the figure and revised the corresponding texts. Figure 13 has turned out to be Figure 14 in the revised manuscript. Please see below:

The maximum values are consistent for the three data sets without regard to the in-situ data having one outlier at Site jldg. This data point is an outlier because the historical weather reports showed no significant snowfall events before or after these dates. This result is a reminder that operational laser measurements of snow depth are not always reliable. For the mean values shown in (b2), the GNSS and in-situ have a better agreement than the PMW because of the significant difference in their spatial footprint. Figure 14 (a3) & (b3) further show the correlation between the GNSS and in-situ or PMW. Accordingly, higher consistencies are achieved from GNSS vs. in-situ than GNSS vs. PMW, with r = 0.75 (RMSD = 4.08 cm) vs. r = 0.57 (RMSD = 6.10 cm) for the maximum and r = 0.90 (RMSD = 1.22 cm) vs. r = 0.75 (RMSD = 3.59 cm) for the mean. The outliers are not involved during the correlations.

[Figure]

**Figure 14.** Site-by-site comparison of the five-year average of the annual maximum /mean snow depth derived from GNSS, in-situ, and PMW, respectively. (a1) The spatial distribution of the sites marked by their corresponding values of the five-year average of the annual maximum snow depth; (b1) Same as (a1) but the annual mean; (a2) The site-by-site comparison of the five-year average of the annual maximum snow depth; (b2) Same as (a2) but the annual mean; (a3) The correlation between the GNSS and in-situ/PMW for the five-year average of the annual maximum; (b3) Same as (a3) but the annual mean. Sixteen sites with the least missing daily snow depth values from 2016 to 2022 are used to draw this figure. The site names are shown in (b2). RMSD: Root Mean Square Difference.

(14) Figure 15 and section 4.3: It is true that the GNSS dataset deliver a higher data rate than the other methods. However due to the lack of reference data at the same rate it is impossible to judge the quality of the 2h dataset. In fact there are several discontinuities in the GNSS derived snow depth (sharp decrease and increase of snow depth) that are normally not seen in snow depth data from a snow storm. It is in fact not possible to judge what changes are due to the real snow depth evolution and which changes are due to artefacts that could be due to i.e GNSS satellite configuration or snow deposited on the GNSS antenna. It is unfortunate that the in-situ snow depth data are not available at

higher rate which is normally possible for laser snow depth instruments.

Colleagues in the China Meteorological Administration tried to find hourly snow measurements again, but unfortunately, they are not available.

We have revised this figure and Section 4.3 to talk about the issue raised by this reviewer. Please see below:

**4.3 Reflection on extreme snow event**

Real-time and accurate monitoring of extreme snow events is of vital practical value. To test if this new GNSS data set can provide supportive information for this application, we use the extreme snow event that happened on February 21 ~ 22 in the year 2015 to analyze the performance of the GNSS, in-situ, and PMW data sets. The event is selected because we have overlapped GNSS data from two GPS/GLONASS compatible sites, i.e., bfqe and bttl, which can provide finer resolution snow depth observations. Figure 16 (a) shows the daily snow depth variations before and after the snow event. As expected, the GNSS and in-situ data have similar responses to the event, while the PMW data has a weak response. These two sites are located in the region with evergreen coniferous forest, which prevents the PMW data from acquiring reliable snow depth values due to its wider observation extent of 25 km. Figure 16 (b) further shows the response of the 2-hour GNSS snow depth data during the week of the event. It captures the evolution of the event in a more detailed way from DOY 51 than that of the other two data sets. However, due to the lack of reference data at the same rate, it is impossible to evaluate the quality of the 2-hour GNSS data set. There are several discontinuities in the GNSS-derived snow depth (i.e., sharp decrease or increase) that are typically not seen in snowstorm data. The common feature of these abnormal values is they all have high STEs (as shown in the bottom panel of Figure 16 (b)). As shown in the top panel of Figure 16 (b), it is possibly due to the relatively low number of tracks used for producing the data set. Regardless of the limitations mentioned above, the GNSS data provides the potential to increase the monitoring frequency of extreme weather in a cheap and effective way in the future, even with a higher resolution of 1-hour or better, particularly for those sites that have compatible observations from more GNSS satellite systems such as GPS, GLONASS, BDS, and Galileo.

[Figure]

**Figure 16.** Performance of the GNSS snow depth on a snow event. (a) Daily data; (b) Two-hour data. Two GPS/GLONASS compatible sites, i.e., bfqe (in red) and bttl (in blue), are used to draw this figure. The error bar of each point in the figure is the standard error (STE) of the snow depths for all the observation records.

(15) Line 550: It is not clear what the authors mean with this sentence.

This sentence no longer exists in the revised manuscript.

(16) Figure 19: I'm missing a yes or no on the diagram arrows indicating in which direction is taken after a decision.

We have added yes/no on the diagram arrows in the figure. Please see below:

[Figure]

**Figure 21.** The strategy of model selection for using GNSS data to retrieve snow depth. CP: carrier phase. Different solutions are represented as Plan A, B, and C.

(17) Conclusion: Please indicate also the (relative) error of the GNSS data compared to the in-situ and PMW data. Not only the internal consistency between the different GNSS systems.

We have added the (relative) error of the GNSS data compared to the in-situ and PMW data in the Conclusion section. The footprints of the three data sets are different. In particular, the PWM is 25 km, which is quite larger compared to the other two. The (relative) errors between GNSS and in-situ and between GNSS and PMW are only for reference and do not represent factual accuracy. We have stated this issue in the previous section, "Section 4.2 Comparison with in-situ measurements and the PMW products: "The GNSS snow depth data set, the PMW data set, and the in-situ measurements are not consistent in terms of the spatial footprint. The GNSS and in-situ data have a closer footprint than the 25-km PMW data. The footprint of GNSS is approximately ~ 30 m x 30 m, as illustrated in the following Figure 17. Due to the discrepancy in footprint, it is impractical to give factual accuracies when comparing these three data sets. Instead, we present the performance of the three data sets at daily scale, multi-year scale, and

interannual variabilities. The RMSD and rRMSD values presented in Figure 13 and Figure 14 are for reference only and do not represent factual accuracies."

Revisions in the "Conclusions" section are as below:

The data set has high internal consistency with regards to different GNSS systems (mean r = 0.98, RMSD = 0.99 cm, and rRMSD = 4.32%), different frequency bands (mean r = 0.95, RMSD = 2.37 cm, and rRMSD =6.60%), and different GNSS receivers (mean r = 0.89). The data set also has high external consistency with the in-situ measurements and the PMW products, with a consistent illustration of the interannual snow depth variability. Results from the 17 GNSS sites with the most extended temporal coverage (i.e., from 2013 to 2022) show better performance between GNSS and in-situ that between GNSS and PMW, with RMSD = 2.08 cm & rRMSD = 10.40% for the former, and RMSD = 3.53 cm & rRMSD = 17.68% for the latter. The results also show the good potential of GNSS to derive hourly snow depth observations for better monitoring snow disasters. The proposed framework to develop the data set provides comprehensive and supportive information for users to process raw data of ground GNSS stations with complex environmental conditions and various observation conditions. The resulting GSnow-CHINA v1.0 data set is distinguished from the current point-scale in-situ data or coarse-gridded data, which can be used as an independent data source for validation purposes. The data set is also useful for regional and global climate research and other meteorological and hydrological applications.

---

## Referee Report (RR1)

The authors addressed all reviewer comments and improved the manuscript quality consistently. However, in my opinion, before the manuscript can be accepted for publication, the part of the manuscript regarding the error analysis must be improved. After these issues are solved, the manuscript should be accepted for publication.

- Section 3.5 is unclear in two points:
a) I don't understand why the authors choose to compute the rRMSE relative to the variation in X , (max(X)-min(X)). For the case min(x)=0 this would be the maximum value (which is the case for this dataset), and if the variation in X is small the relative values diverge. I would use either the mean value as for the STE, or normalize each error (Y-X) by X before computing the root mean. Please change it or give a valid reason for using this type of normalization.
b) It is also not clear from the text which dataset Z is used for determining the STE. I assume this are the different traces used for one datapoint. Please explain more it in detail. Since the STE is given for each sample should it not be more correct to use the standard deviation? Moreover, if Z is the sample of traces used for one datapoint then N must be the number of traces, which is different of the N used for RMSE which is the number of datapoints. Use different symbols if this is the case.

- I don't agree with the argument of the authors that it is not necessary to indicate the RMSE and rRMSE of the data in figure 11. It is true that comparison between in-situ and GNSS data is tricky due to the different footprint. However, some information is better than no information. The footprint point can still be explained in the text. Please indicate RMSE and rRMSE for figure 11 a1,b1 and c1. The same is true for sentence at line 510.
Moreover, the second part of figure 11 is visually very appealing but unfortunately the higher bars cover the points in the back. I suggest using a simple color plot since the z-axis is already indicated by the color. (At least I think since the z-axis label is missing). I also find valuable to use a color plot similar to Figurea1,b1 also for fig. 11a1,b,c1. It would indicate the density of point relative to the 1:1 line.

**Below a few minor comments.** Mostly style and language. The line numbers refer to the document with track changes.

Line 40: delete good. "the potential" is sufficient.

Line 47: delete "years of"

Lines 68-69: ''remote sensing has a long revisiting period (>20 days) and high cost …"

Line 99: " … for typical GNSS-IR sites…"

Line 128: It is not clear to me in which sense snow is a natural disaster for pastural area if is providing fresh water. Do you mean the lack of snow?

Figure 4: Please indicate E in (a).

Line 285: Indicate the return period. Do the tracks repeat every day?

Line 303: housing instead of small house.

Line 336: It is not clear to which site you are referring to. Figure 5 has 3 sites.

Line 411: "…site over from October…". Delete ''over''.

Line 421: included instead of involved.

Line: 424: Please change to "… possible error due to vegetation…"

 Line 474: See comment above about STE. Do you indicate the same STE for each point?

Lines 475-476: It is not clear to me what the authors mean with this sentence. Did you filter all data with large STE. In this case why? Specify how many outliers (percentage) were deleted and precise which other effects besides GNSS frequency you expect.

Lines 496-500: It is not cleat to me relative to which data the RMSE and rRMSE indicated in this paragraph were computed. Please specify.

Lines 518-521: All snow depth values? I see just 16 sites but the dataset contains much more sites. Please correct or specify.

Figure 14 and corresponding text: Can you give an explanation for the underestimation of mean snow depth by PMW? It has a larger footprint but why is the mean snow depth over a larger footprint smaller?

Line 578: Change to "PMW data were available only for the period 2016-2020."

Line 689: Provide a reference of the precious study.

Line 693: change to ''30 seconds for which is impossible …"

---

## Author Response (AR2)

**Response to Topical Editor Dr. Xin Li:**

Comments to the author:

The manuscript has been greatly improved, however, it still needs to revise some parts, please refer to the reviewer's comments for further revision.

We thank Topical Editor Dr. Xin Li for handling this manuscript. We have carefully revised the manuscript to address the issues and comments raised by the reviewer during this round. In particular, we have revised the figures and texts related to the error analysis. We have also corrected the style and language issues raised by the reviewer. Please see detailed information in the following point-to-point responses.

**Response to the Reviewer:**

(1) The authors addressed all reviewer comments and improved the manuscript quality consistently. However, in my opinion, before the manuscript can be accepted for publication, the part of the manuscript regarding the error analysis must be improved. After these issues are solved, the manuscript should be accepted for publication.

We thank this reviewer for her/his valuable time in reviewing this manuscript again and for providing thorough and insightful comments. We have carefully revised the manuscript to address the issues and comments raised during this round. In particular, we have revised the figures and texts related to the error analysis. We have also corrected the style and language issues raised by the reviewer.

Point-to-point responses are listed below. Comments are in **black**, the authors' responses are in **blue**, and the revisions in the manuscript are in **red**.

- Section 3.5 is unclear in two points:

(2) a) I don't understand why the authors choose to compute the rRMSE relative to the variation in X, (max(X)-min(X)). For the case min(x)=0 this would be the maximum value (which is the case for this dataset), and if the variation in X is small the relative values diverge. I would use either the mean value as for the STE, or normalize each error (Y-X) by X before computing the root mean. Please change it or give a valid reason for using this type of normalization.

We have changed the relative RMSD (rRMSD) to normalized RMSD (nRMSD), and the relative STE (rSTE) to normalized STE (nSTE) for simplification. We have revised to use the mean value to normalize the RMSD. Please see below for the revised texts:

The Root Mean Square Difference (RMSD), normalized RMSD (nRMSD), STE, and normalized STE (nSTE) are four error indicators used in this study. The RMSD of two data (X and Y) are given by $RMSD = \sqrt{\sum (X_i - Y_i)^2 / N}$, where N is the number of elements in the sample. The nRMSD is given by $nRMSD = RMSD/(\text{mean}(X))$. The STE of one data (Z) is given by $STE = \sigma_Z / \sqrt{N_Z}$, where $\sigma_Z$ is the standard deviation of the data Z and $N_Z$ is the number of elements in Z. The nSTE is given by $nSTE = STE/\bar{Z}$, where $\bar{Z}$ is the mean of the sample.

The nRMSD normalized by the mean value is larger than by the (Max-Min). We have added several sentences in the revised Section 4.1 to explain this issue:

"… We show the correlation coefficient (r), RMSD, and nRMSD values for each comparison. It should be noted that the nRMSD (snow depth > 5 cm) is significantly lower than the nRMSD (all), which is because the reference value (i.e., the mean snow depth) was used to normalize the RMSD. A large portion of snow depths in the study area is lower than 5 cm, yielding a lower mean value when involving all the data than only using the > 5 cm data. The same principle applies to the following Figures 9, 10, and 11. Nevertheless, the metrics only represent the comparison during the intermediate process of the data set production. Users can define their own rules to use the data according to the quality flags in the published data set."

(3) b) It is also not clear from the text which dataset Z is used for determining the STE. I assume this are the different traces used for one datapoint. Please explain more it in detail. Since the STE is given for each sample should it not be more correct to use the standard deviation? Moreover, if Z is the sample of traces used for one datapoint then N must be the number of traces, which is different of the N used for RMSE which is the number of datapoints. Use different symbols if this is the case.

We apologize for the unclear description of the STE. Yes, the different traces used for one data point are used to determine the STE. We have added explanations in the titles of Figures 9, 16, and 19 to clarify this issue:

"… The error bar of each point is the standard error (STE) of the snow depths for all the available tracks of this point. …"

We have revised the expression of the *STE* to replace *N* with $N_Z$ to distinguish these two numbers. Please see below:

"…The STE of one data (Z) is given by $STE = \sigma_Z/\sqrt{N_Z}$, where $\sigma_Z$ is the standard deviation of the data Z and $N_Z$ is the number of elements in Z. …"

We agree with the reviewer that the standard deviation (STD) is also a helpful indicator. In this study, we use the STE instead of STD, the same as that used in the published west U.S. PBO $H_2O$ GNSS-IR snow depth data set on the National Snow and Ice Data Center (NSIDC, https://nsidc.org).

Reference:

Larson, K. M. and E. E. Small. 2017. *Daily Snow Depth and SWE from GPS Signal-to-Noise Ratios, Version 1*. Boulder, Colorado USA. NASA National Snow and Ice Data Center Distributed Active Archive Center. doi: https://doi.org/10.5067/Z02Y1HGNFXCH.

(4) - I don't agree with the argument of the authors that it is not necessary to indicate the RMSE and rRMSE of the data in figure 11. It is true that comparison between in-situ and GNSS data is tricky due to the different footprint. However, some information is better than no information. The footprint point can still be explained in the text. Please indicate RMSE and rRMSE for figure 11 a1,b1 and c1. The same is true for sentence at line 510.

We have added the RMSD and nRMSD in Figure 11 a1, b1, and c1. Please see the revised figure in the following Comment # (5). We have also added a sentence to explain the footprint issue:

"Due to the inconsistent footprint between the GNSS and in-situ measurements, the error metrics presented in Figure 11 are for reference only and do not represent factual accuracies."

(5) Moreover, the second part of figure 11 is visually very appealing but unfortunately the higher bars cover the points in the back. I suggest using a simple color plot since the z-axis is already indicated by the color. (At least I think since the z-axis label is missing). I also find valuable to use a color plot similar to Figurea1,b1 also for fig. 11a1,b,c1. It would indicate the density of point relative to the 1:1 line.

We have remade Figure 11 a2, b2, and c2 using the display style of "tile". We have also changed to use density plot for (a1), (b1), and (c1). To keep consistence with the previous Figures 9 and 10, we changed to use the > 5 cm data to plot (a1), (b1), and (c1). The revised figure 11 is shown below:

[Figure]

**Figure 11.** Comparisons of the GNSS-derived snow depth and the in-situ measurements from different types of GNSS receivers: (a1) Trimble; (b1) Leica; (c1) Minshida (MSD), and the histogram of the standard error (STE) and nSTE of snow depths for different types of GNSS receivers: (a2) Trimble; (b2) Leica; (c2) MSD. The number of sites representing Trimble, Leica, and MSD is 20, 5, and 24. The GNSS snow depths values are greater than 5 cm in this figure. To prevent other possible effects besides the receiver type, the STE of snow depths is less than 1 cm (63% of the entire data) in (a1), (b1), and (c1). RMSD: Root Mean Square Difference; nRMSD: normalized RMSD.

(6) Below a few minor comments. Mostly style and language. The line numbers refer to the document with track changes.

Line 40: delete good. "the potential" is sufficient.

Fixed.

Line 47: delete "years of"

Fixed.

Lines 68-69: ''remote sensing has a long revisiting period (>20 days) and high cost …"

Fixed.

Line 99: " … for typical GNSS-IR sites…"

Fixed.

Line 128: It is not clear to me in which sense snow is a natural disaster for pastural area if is providing fresh water. Do you mean the lack of snow?

Sorry for the unclear description. We have revised this sentence as below:

Northern China lies between latitudes 25°N and 55°N and longitudes 70°E and 140°E and includes humid, semi-humid, semi-arid, and arid zones. Snow is the primary freshwater resource  in this area . Sudden snowstorms or long-lasting deep snow is one of the major natural disasters for pastoral areas because it affects livestock grazing.

Figure 4: Please indicate E in (a).

We have added E in (a). Please see below:

[Figure]

Line 285: Indicate the return period. Do the tracks repeat every day?

We have clarified this issue. Please see below:

It is worth mentioning that GPS ground tracks have sidereal repeatability and reappear at the same azimuth every day.

Line 303: housing instead of small house.

Fixed.

Line 336: It is not clear to which site you are referring to. Figure 5 has 3 sites.

We have clarified this issue. Please see below:

At the same time, it is impossible to derive correct $h_0$ values for "bgfc" in other orientations that have buildings or trees; This phenomenon can be verified from the photo of the site in Figure 5.

Line 411: "…site over from October…". Delete ''over''.

Fixed.

Line 421: included instead of involved.

Fixed.

Line: 424: Please change to "… possible error due to vegetation…"

Fixed.

Line 474: See comment above about STE. Do you indicate the same STE for each point?

For one specific point, the different traces are used to determine the STE of this point. Therefore, each point has its own STE value. Please see detailed responses in the previous Comment # (3).

Lines 475-476: It is not clear to me what the authors mean with this sentence. Did you filter all data with large STE. In this case why? Specify how many outliers (percentage) were deleted and precise which other effects besides GNSS frequency you expect.

We filtered out the data with large STE (i.e., STE > 1 cm). We have specified the percentage of valid data in the titles of Figures 9, 10, and 11.

The 1 cm is a rigorous threshold. The manuscript states that GNSS constellation, frequency, and receiver type are key factors affecting snow depth accuracy. If we compare one of the three factors, we should prevent the other two and other random errors from cross-influence. In other words, we should ensure a snow depth value is "accurate" under the defined condition. Therefore, we use the STE = 1 cm threshold to prevent this issue.

We have added explanations in the revised manuscript Section 4.1:

"The intra-comparisons of the snow depths are executed from three aspects, i.e., comparison of different GNSS constellations, frequency bands, and receivers. If we compare one of the three factors, we should prevent the other two and other random errors from cross-influence. In other words, we should ensure a snow depth value is "accurate" under the defined condition. Therefore, in this section, we use a rigorous threshold of STE = 1 cm to filter out the outliers. …"

We have also remade Figures 9 and 10 considering the reviewer's comments comprehensively, i.e., we revised (or added) the nRMSD, revised the style of the density plot, and used a consistent 1 cm threshold for all the figures. For the convenience of the reviewer, the revised Figures 9 and 10 are shown below:

[Figure]

**Figure 9.** Correlations of 24 h/12 h snow depths from GPS and GLONASS observations. (a) 24 h; (b) 12 h. The error bar of each point is the standard error (STE) of the snow depths for all the available tracks of this point. Four available sites, i.e., hltl, hlhl, bfqe, and bttl, during the GPS/GLONASS overlapped periods (i.e., the year 2014 and 2015) are used to plot this figure. For each point in the figure, the number of valid observations is more than five. To prevent other possible effects besides the GNSS constellation, the STE of snow depths is less than 1 cm (90% for the 24 h data and 76% for the 12 h data). Blue points are with the retrieved GPS and GLONASS snow depths greater than 5 cm. RMSD: Root Mean Square Difference; nRMSD: normalized RMSD.

[Figure]

**Figure 10.** Correlations of snow depth from different GNSS frequencies. (a1) GPS L1 vs. GPS L2; (b1) GLONASS L1 vs. GLONASS L2. The color bar represents the density of points; (a2) Same as (a1) but with snow depths greater than 5 cm; (b2) Same as (b1) but with snow depths greater than 5 cm. Fifty-one high-quality GPS sites of CMA and four GPS/GLONASS compatible sites are respectively used to plot (a1, a2) and (b1, b2). For each point in the figure, the number of valid observations is more than five. To prevent other possible effects besides the GNSS frequency, the STE of each snow depth is less than

1 cm in all the subfigures (61% for the GPS data and 70% for the GLONASS data). RMSD: Root Mean Square Difference; nRMSD: normalized RMSD.

Lines 496-500: It is not cleat to me relative to which data the RMSE and rRMSE indicated in this paragraph were computed. Please specify.

We apologize for the unclear description. The r, RMSD, and nRMSD values are the average values from Figures 9 ~ 11. For example, in Figure 9, the RMSD of the 24-hour and 12-hour GPS/GLONASS results is 1.01 cm and 0.97 cm respectively, then the mean RMSD is 0.99 cm. We have revised the corresponding texts to clarify this issue:

"From the comprehensive intra-comparisons shown in Figures 9 ~ 11, we conclude that the snow depths derived from different GNSS constellations, frequency bands, and receivers have overall good agreement. The average values of the metrics shown in Figures 9 ~ 11 are summarized as follows: mean r = 0.98, mean RMSD = 0.99 cm, and mean nRMSD (snow depth >5 cm) = 0.11 for different GNSS constellations, mean r = 0.97, mean RMSD = 1.46 cm, and mean nRMSD (snow depth >5 cm) =0.16 for different frequency bands, and mean r = 0.62 for different GNSS receivers. Therefore, it is feasible to combine all these results to produce the snow depth data set in this study."

Lines 518-521: All snow depth values? I see just 16 sites but the dataset contains much more sites. Please correct or specify.

We have clarified this issue. Please see below:

Figure 12 presents all the GNSS snow depth values  of the 16 GNSS sites, regardless of its quality, to give a comprehensive illustration of the data.

Figure 14 and corresponding text: Can you give an explanation for the underestimation of mean snow depth by PMW? It has a larger footprint but why is the mean snow depth over a larger footprint smaller?

We have added explanations for this issue. Please see below:

For the mean values shown in (b2), the GNSS and in-situ have a better agreement than the PMW because of the significant difference in their spatial footprint. Most sites are located in the region with evergreen coniferous forest, which prevents the PMW data from acquiring reliable snow depth values due to its wider observation extent of 25 km.

Line 578: Change to "PMW data were available only for the period 2016-2020."

Fixed.

Line 689: Provide a reference of the precious study.

We have added the reference:

Wan, W., Larson, K. M., Small, E. E., Chew, C. C., and Braun, J. J.: Using geodetic GPS receivers to measure vegetation water content, GPS Solutions, 19, 237-248, 10.1007/s10291-014-0383-7, 2015.

Line 693: change to ''30 seconds for which is impossible …"

Fixed.

---

## Author Response (AR3)

**Response to Topical Editor Dr. Xin Li:**

We thank Topical Editor Dr. Xin Li again for handling this manuscript. We have revised the manuscript according to the editor's comments. Please see detailed information in the following point-to-point responses.

There are several comments as follows:

1. Since you have put the data at the National Tibetan Plateau/Third Pole Environment Data Center, you are welcome to cite the relevant introduction papers into the articles as: https://doi.org/10.1175/BAMS-D-21-0004.1 and https://doi.org/10.1175/BAMS-D-19-0280.1

Response: We have added citations in "Section 1 Introduction":

Currently, snow cover products derived from optical remote sensing data present high accuracy (Hao et al., 2021), but snow depth products show significant uncertainties. …

…Previous studies also demonstrated that current snow depth data sets and snow water equivalent data sets show significant inconsistencies and uncertainties, which limit their applications in climate change projections and hydrological processes simulations (Xiao et al., 2020; Zhang et al., 2021; Shao et al., 2022)….

2. Since the research is related to snow, it is suggested that the author may reference relevant articles in the special issue,such as https://doi.org/10.5194/essd-13-4711-2021, and https://doi.org/10.5194/essd-14-795-2022

Response: We have added citations in "Section 7 Data availability":

The GSnow-CHINA v1.0 data set is archived and available at National Tibetan Plateau/Third Pole Environment Data Center (Li et al., 2020; Pan et al., 2021) via https://doi.org/10.11888/Cryos.tpdc.271839 (Wan et al., 2021).